# CORRECTING FLAWS IN COMMON DISENTANGLEMENT METRICS

## ABSTRACT

Recent years have seen growing interest in learning disentangled representations, in which distinct features, such as size or shape, are represented by distinct neurons. Quantifying the extent to which a given representation is disentangled is not straightforward; multiple metrics have been proposed. In this paper, we identify two failings of existing metrics, which mean they can assign a high score to a model which is still entangled, and we propose two new metrics, which redress these problems. First, we use hypothetical toy examples to demonstrate the failure modes we identify for existing metrics. Then, we show that similar situations occur in practice. Finally, we validate our metrics on the downstream task of compositional generalization. We measure the performance of six existing disentanglement models on this downstream compositional generalization task, and show that performance is (a) generally quite poor, (b) correlated, to varying degrees, with most disentanglement metrics, and (c) most strongly correlated with our newly proposed metrics. Anonymous code to reproduce our results is available at https://github.com/anon296/anon.

## 1 INTRODUCTION

Early proponents of neural networks argued that a significant advantage was their ability to form distributed representations, where each input is represented by multiple neurons, and each neuron is involved in the representation of multiple different inputs (Hinton et al., 1986). Compared to using a separate neuron for each input, distributed representations are exponentially more compact (Bengio, 2009). An extension of distributed representations is the idea of disentangled representations, where each neuron represents a single human-interpretable feature of the input data, such as colour, size or shape. These features are often referred to as "generative factors" or "factors of variation". Intuitively, a disentangled vector representation is one in which a certain subset of neurons represents (for example) shape and shape only, another distinct subset represents size and size only, etc., and changing the size of the input but not the shape, will mean that the size neurons change their activation value, but the shape neurons remain unchanged. In the strongest case, each factor is represented by a single neuron so that, e.g., changing the colour of the object in the image would cause a single neuron to change its value while all other neurons remain unchanged. (We discuss further below the ambiguity as to whether this stronger condition is required.) Disentanglement (DE) was originally formulated by Bengio (2009) (see also Bengio et al. (2013); Bengio (2013)). More recently, following Higgins et al. (2016), there have been many unsupervised DE methods proposed based on autoencoders.

In this paper, we examine the commonly used metrics to assess disentanglement. Firstly, we show how they fail to pick up certain forms of entanglement, and that representations can score highly on such metrics while being entangled. Specifically, we expose two problems with existing metrics: that they incorrectly align ground-truth factors to neurons, as they do not require distinct variables to be assigned to distinct factors; and that they measure the strength of the relationship of features to individual neurons, which is fundamentally different from the relationship to sets of neurons, and hence can give undesirable results. We show that these problems occur in practice by examining the results of six different DE models trained on three different datasets.

To address these problems, we present two new DE metrics, based on the ability of a classifier to predict the generative factors from the encoded representation. If a representation is truly disentangled, then all the relevant information should be contained in a single neuron (or possibly a few neurons,

see discussion in Section 2), and so a classifier using only this/these neuron(s) should be just as accurate as one using all neurons, and one using all other neurons should be very inaccurate. Our first metric is the accuracy of the single-neuron classifier. Our second metric is the difference between the accuracy of a classifier using all neurons, vs one using all neurons but the single selected neuron.

We also establish the superiority of our proposed metrics using a downstream compositional generalization task of identifying novel combinations of familiar features. Humans could recognize a purple giraffe, even if we have never seen one or even heard the phrase "purple giraffe" before, because we have disentangled the concepts of colour and shape, so could recognize each separately. The ability to form and understand novel combinations is a deep, important aspect of human cognition and is a direct consequence of humans being able to disentangle the relevant features of the objects we encounter. Our downstream task tests, for example, whether a network trained to identify blue squares, blue circles and yellow squares can, at test time, correctly identify yellow circles. If it had learned to disentangle colour from shape, then it could simply identify "yellow" and "circle" separately, each of which is familiar. We show that existing DE models generally perform poorly at this task, suggesting they are further from DE than previous analyses have implied. We also show that a high score on DE metrics is predictive of performance on this task, and that our proposed DE metrics are the most predictive in this respect. Our contributions are briefly summarized as follows.

- We identify and describe two shortcomings of existing DE metrics: (1) incorrect alignment of neurons to factors and (2) focusing on the importance of individual neurons instead of sets of neurons.
- We show, experimentally, that these problems with existing metrics occur in practice.
- We propose two alternative metrics, (1) single-neuron classification and (2) neuron knockout, that do not suffer from the problems that existing metrics suffer from.
- We validate our metrics on a downstream task of compositional generalization. We show empirically that, while existing models generally perform badly at recognizing novel combinations of familiar features (compositional generalization), their performance correlates with DE metrics, and correlates most strongly with our proposed metrics.

The rest of this paper is organized as follows. Section 2 discusses related work. Section 3 describes the shortcomings of existing DE metrics. Section 4 proposes our new metrics. Section 5 presents empirical results, including those on the downstream compositional generalization task. Section 6 then discusses some limitations and future work, and summarizes our contributions.

## 2 RELATED WORK

**Disentanglement Models.** After the initial proposal by Bengio (2009), disentangled representations received new interest beginning when Higgins et al. (2016) proposed $\beta$-VAE, as an adaption of the variational autoencoder (VAE) (Kingma and Welling, 2013). By taking the prior to have a diagonal covariance matrix, and increasing the Kullback-Leibler divergence (KL) loss weight, $\beta$-VAE encourages the model representations to have diagonal covariance too, which the authors claim enforces DE. Kumar et al. (2017) further encouraged a diagonal covariance matrix by minimizing the Euclidean distance of the model's covariance matrix from the identity matrix. Burgess et al. (2018) proposed to gradually increase the reconstruction capacity of the autoencoder by annealing the KL in the VAE loss. Note that these works implicitly equate uncorrelated variables (i.e., diagonal covariance matrix) with independent variables, which is incorrect. For example, $Y = X^2$ are totally uncorrelated but totally dependent. Chen et al. (2018) propose $\beta$-TCVAE, which minimizes the total correlation of the latent variables, approximated using Monte-Carlo based on importance sampling. Kim and Mnih (2018) propose FactorVAE, which also minimizes total correlation, this time approximating using a discriminator network. Locatello et al. (2019) challenged earlier DE methods, proving that it is always possible for a model to learn an entangled representation that appears disentangled only on the available data, and presenting experiments that call into question whether disentangled representations lead to superior downstream performance, as previous works had claimed. Later, Locatello et al. (2020) claimed that including a small amount of supervision was sufficient to learn disentangled representations. More recently, other authors have pushed back against the claim that unsupervised DE is impossible, arguing that the priors embodied in autoencoder architectures still allow unsupervised DE in practice Roth et al. (2022); Rolinek et al. (2019). The models we test on

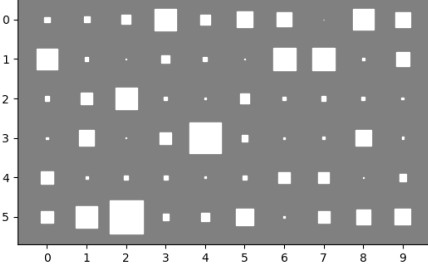

| | blue | yellow |
|---|---|---|
| Square | 0 | 0 or 1 with equal probability |
| Square | 0 or 1 with equal probability | 1 |

Figure 1: Left: Hinton diagram for $\beta$-TCVAE on 3dshapes. The size of the square at $(i, j)$ shows the mutual information between factor $i$ and neuron $j$. Existing metrics incorrectly align both factor 2 and factor 5 to neuron 2, whereas we correctly align factor 2 to neuron 1 instead. Right: Toy example of partially encoding two factors.

include a range of both unsupervised and weakly supervised. Recent unsupervised models include Klindt et al. (2020), a VAE-based model to learn disentangled representations from videos of natural scenes. Semi-supervised methods include WeakDE Valenti and Bacciu (2022), which adversarially pushes the latent distribution for each factor close to a prior computed using a small labelled subset. Some DE models are fully supervised, such as MTD (Sha and Lukasiewicz, 2021), which partitions the neurons into a subset for each factor, and defines several losses using the ground-truth labels. Some can operate either supervised or unsupervised, such as Parted-VAE (Hajimiri et al., 2021), which minimizes the Bhattacharyya distance (Bhattacharyya, 1946) from a multivariate normal.

**Disentanglement Metrics.** Higgins et al. (2016) made an early attempt to quantify disentanglement, by fixing one generative factor and varying others, then predicting which factor was fixed from the mean each absolute difference. Kim and Mnih (2018) replace mean absolute difference with the index of the lowest variance neuron. Ridgeway and Mozer (2018) decompose DE into modularity and explicitness, each with its own metric. They measure modularity as the deviation from an "ideally modular" representation, where every latent neuron has nonzero mutual information with exactly one generative factor. Explicitness is measured similarly to the metric of Kim and Mnih (2018), except using the mean of one-vs-rest classification and AUC-ROC. The SAP score Kumar et al. (2017) calculates, for each factor, the $R^2$ coefficient with each latent dimension, and then takes the difference between the largest and second largest. A representation will score highly on SAP if, for each generative factor, one neuron is very informative and no other individual neuron is. A similar idea is employed by the mutual information gap (MIG) metric (Chen et al., 2018), except using mutual information instead of $R^2$. Eastwood and Williams (2018) decompose the task into disentanglement, completeness and informativeness (DCI). Then, they train a classifier (commonly a linear model or a gradient-boosted tree) to predict each generative factor from the latent factors, and estimate DCI from a measure of importance of each feature to each factor from the classifier. IRS (Suter et al., 2019) measures the maximum amount that a given neuron can be changed by changing a generative factor other than the one it corresponds to. Another recent metric is MED Cao et al. (2022), which is the same as DCI, using mutual information as feature importance.

**Definitions of Disentanglement.** There is ambiguity in the literature as to whether DE requires that each factor is represented by a single neuron (strong DE), or allows representation by multiple neurons (weak DE). The original definition by Bengio (2009) (echoed by Higgins et al. (2016), Kim and Mnih (2018) and others) stipulates that "each neuron represents a single factor", which seems to allow that each factor is represented by multiple neurons. Other treatments of DE imply a stronger notion of the concept, namely, an injective function from factors to the unique neurons that represent them. This is implicit in the common metrics that map each factor to a single neuron, in the technique of latent traversals that aim to vary a single factor through its range of values by varying a single neuron, and in the descriptions of DE in other works, e.g., "learning one exclusive factor per dimension" (Pineau and Lelarge, 2018). Attempts at formal definitions, e.g. equivariance in group theory (Higgins et al., 2018), allow the possibility of many-to-one mappings in theory, but present examples and discuss benefits of DE with respect to one-to-one mappings. Weak DE is less compact than strong DE, and loses some interpretability benefits: it would be hard to tell which neurons represent a given factor if we have to check all *subsets* of neurons, of which there

are exponentially many. Our proposed metrics focus on the strong notion, for the following reasons: (1) strong DE is generally assumed by many common DE metrics, (2) strong DE is assumed by the common practice of latent traversals, (3) weak DE loses interpretability as compared with strong DE. This does not mean that weak DE is without value, indeed some authors have argued that it is more suitable for certain factors to be represented by multiple neurons Esmaeili et al. (2022). However, it is not possible for a single metric to measure both because they are different objectives. For example, if each factor is perfectly represented by a distinct set of 5 neurons, which don't overlap, then the representation shows perfect weak DE and should get a max score by a metric measuring weak DE. But it could be that none of these 5 neurons individually represent the corresponding factor (see Section 3.2), in which case the representation shows poor strong DE and should get a low score by a metric measuring strong DE. Thus, strong DE is valuable to measure because it has some advantages that weak DE does not (compactness, interpretability), and, as we have just argued, it requires a metric specifically dedicated to strong DE; there cannot be a metric that adequately measures both strong and weak DE at the same time. Therefore, there is value to a metric for strong DE, and that is what we provide in this paper.

## 3 PROBLEMS WITH EXISTING DISENTANGLEMENT METRICS

### 3.1 INCORRECT ALIGNMENT OF LATENT VARIABLES

The majority of existing metrics are based on aligning the set of factors $G$ with the set of neurons $Z$; that is, for each factor, finding the neuron that it is represented by. Each neuron is only supposed to represent a single factor, however, existing metrics simply relate each factor to the maximally informative variable, e.g., as measured by mutual information (Chen et al., 2018) or weight from a linear classifier (Kumar et al., 2017). This does not enforce the constraint of having distinct neurons for distinct features, it means that the same neuron could be selected as representing multiple different factors. For example, consider again the model trained on a dataset of blue squares, blue circles, yellow squares and yellow circles. Suppose such a model has two latent variables, $z_1$ and $z_2$, the first is as shown on the right of Figure 1, and the second is random noise, unrelated to the inputs. Then $z_1$ encodes both colour and shape, each to an accuracy of 75%, whereas $z_2$ encodes both only to 50% (random guess). Thus, $z_1$ will be chosen as the representative of both colour and shape.

This situation is not merely theoretical, it often occurs in practice. One example is the left of Figure 1, which shows a Hinton diagram for $\beta$-TCVAE trained on 3dshapes (further examples of other datasets and models are given in the appendix). There, the existing approach of aligning each factor to the most informative neuron (e.g., by mutual information), incorrectly concludes that two different factors, size (factor 2) and colour (factor 5), should be aligned to the same neuron (neuron 2). In a Hinton diagram, the size of the square is proportional to the MI, and the square at (5,2) (using matrix indexing) the biggest in its row, so existing metrics align factor 5 to neuron 2. The square at (2,2) is also the biggest in its row, so existing metrics also align factor 2 to neuron 2. This is not good, because each neuron should only be aligned with one factor. Our method enforces aligning distinct factors to distinct neurons and so avoids this problem. It ends up aligning factor 2 to neuron 1 and factor 5 to neuron 2.

The model from the right of Figure 1 (call it M1), would get a higher score than another (call it M2) in which $z_1$ is as above and $z_2$ represents shape to an accuracy of $70\%$. The appendix gives calculations of MIG, DCI and SAP in this case, which all give a higher score to M1 than M2, and our metrics, which do the opposite. What our metrics do is incorrect, as M2 is closer to the desired case where $z_1$ represents colour and colour only, and $z_2$ represents shape and shape only. M1 needs $z_2$ to learn shape and $z_1$ to unlearn shape, M2 just needs $z_1$ to unlearn shape. See also the appendix in Cao et al. (2022), which mentions a similar failing.

### 3.2 CONFLATING REPRESENTATION BY INDIVIDUAL NEURONS VS BY A SET OF NEURONS

The second problem with existing metrics is how they handle information distributed over multiple neurons. Let $g_0, \ldots, g_n$ be the ground-truth generative factors, and $z = z_0, \ldots, z_m$, $m \geq n$ the corresponding neurons and suppose $g_1$ has been aligned to $z_1$. Further, let $z_{\neq i}$ be the set of neurons other than $z_i$. Now, consider XOR: $g_1 = z_2 \oplus z_3$. (We use discrete values for clarity, a continuous approximation could be $|z_2 - z_3|, z_2, z_3 \in [0, 1]$.) Almost all existing metrics, would fail here. Those based on linear classifiers Kumar et al. (2017); Ridgeway and Mozer (2018); Eastwood and Williams

(2018) would conclude that $z_2$ and $z_3$ are unrelated to $g_1$, because the strength of the linear relationship in XOR is zero. Those that use pairwise mutual information Chen et al. (2018); Cao et al. (2022); Do and Tran (2019) would also react incorrectly. They would find that $I(g_1; z_2) = I(g_1; z_3) = 0$, and so conclude that $g_1$ is not represented by $z_{\neq i}$, when in fact $g_1$ is perfectly represented by $z_{\neq i}$. In Section 5.1 we show these distributed entanglements occur in practice.

DCI can be used with a non-linear classifier, most commonly a gradient-boosted tree, but it still computes an importance score for each feature individually to each factor, so fails for a different but similar reason. This is shown by Theorem 3.1, using an extension of XOR to multiple feature values.

**Theorem 3.1.** *As the number of neurons and factors increases, the $D$ and $C$ components of DCI can, under a very broad class of feature importance measures, including gradient-boosted trees, assign a score that is arbitrarily close to perfect, even though the model is completely entangled in the sense that no neuron, by itself, contains any information about any generative factor.*

The full proof is in the appendix. Intuitively, the proof shows that it is possible for $n$ neurons to perfectly encode $n$ factors, where each factor depends only on two neurons, but where knowing the value of any one neuron reveals nothing about the value of any factor. It is proved in the appendix for the case of uniformly distributed feature values. This situation constitutes a false positive for DCI. The representation is not disentangled at all, but DCI gives it a high score. A good metric should function in both uniform and non-uniform cases, so this demonstrates a shortcoming of DCI. The only assumption on the feature importance measure is that it assigns a score of 0 whenever the mutual information, even when conditioned on any number of other neurons, is zero. This is a weak assumption and should be met by any reasonable feature importance measure. We do not need very many neurons/factors before there can be occasions on which the value of D and C becomes spuriously high. For 10 neurons/factors, $D, C \geq 0.8$; for 20, $D, C \geq 0.9$. While DCI to some extent approximates measuring both weak DE and strong DE, Theorem 3.1 shows that it can be an unreliable measure of strong DE, further motivating the search for reliable strong DE metrics.

## 4 PROPOSED METRICS

**Single-neuron Classification (SNC)**    Our first metric, SNC, begins by aligning factors to neurons. Unlike prior works, we align all factors simultaneously so that we can enforce aligning distinct factors to distinct neurons:

$$\arg\max_{\{f:G\to Z\,|\,\text{f is injective}\}} \sum_{g\in G} I(g; f(g)), \tag{1}$$

where $I$ denotes mutual information (this could be replaced with $R^2$ or any other measure of informativeness). A solution to equation 1 can be computed efficiently using the Kuhn-Munkres algorithm (Munkres, 1957), giving a mapping in which no two factors are mapped to the same neuron. This better fits the notion of DE than previous approaches. After alignment, we use $z_i$ as a classifier, by dividing its values across the dataset into bins, where the bin size is the greatest common divisor of the size of classes in the dataset. We then align these bins with the $K$ ground-truth classes, and compute the chance-adjusted accuracy Let $X = (x)_{1 \leq i \leq N}$ denote the data, $c : X \to \{0, \dots, K-1\}$ specify the ground truth labels, and and $b : X \to \{0, \dots, K-1\}$ specify the bin index after alignment. Then, the metric score is the accuracy $a$ on a single neuron, adjusted for chance accuracy $r$:

$$SNC = \max(0, \frac{a - r}{1 - r}),$$

where $a = \frac{1}{N} \sum_{i=1}^{N} \mathbb{1}(b(x_i) = c(x_i))$ and $r = \frac{1}{N^2} \sum_{j=1}^{K} (\sum_{i=1}^{N} \mathbb{1}(c(x_i) = j)^2)$. This essentially quantifies the property that latent traversals aim to show qualitatively. In the terminology of Ridgeway and Mozer (2018), it is a measure of explicitness, except that they fit a linear classifier on all neurons, not just $z_i$. That is, if, for each $g_i$, there is some line in representation space such that the representation vector encodes $g_i$ as the distance of the projection along that line, then this is regarded as disentangled. Using a single-neuron classifier, on the other hand, requires that line to be an axis. In the appendix, we also show linear classifier scores, which are broadly similar.

**Neuron Knockout (NK)**    Our second proposed metric is inspired by the technique of gene knockout in genetics, which tests how relevant a given gene is to a given function, by removing the gene and

|  |  | $\beta$-VAE | $\beta$-TCVAE | FactorVAE | PartedVAE | PartedVAE-ss | weakde |
|---|---|---|---|---|---|---|---|
| Dsprites | SNC | 24.8 (3.20) | 41.3 (3.00) | 15.1 (1.78) | 16.1 (4.54) | 19.0 (4.80) | 5.8 (0.75) |
|  | MLP | **89.4 (3.95)** | **86.2 (1.10)** | **77.1 (4.10)** | **70.7 (10.33)** | **48.5 (9.08)** | **99.8 (3.50)** |
|  | NK | 39.9 (1.11) | 31.3 (4.10) | 32.9 (1.45) | 34.9 (4.23) | 14.5 (8.17) | 5.9 (1.17) |
| 3dshapes | SNC | 19.9 (4.70) | 20.1 (2.06) | 16.9 (3.36) | 43.8 (24.97) | 68.3 (9.55) | 15.4 (2.11) |
|  | MLP | **99.8 (3.50)** | **99.9 (2.50)** | **98.1 (7.24)** | **96.2 (6.83)** | **88.9 (14.53)** | **99.8 (0.68)** |
|  | NK | 8.0 (0.20) | 8.1 (0.11) | 16.8 (2.94) | 60.2 (6.19) | 41.8 (8.07) | 2.5 (0.18) |
| MPI3D | SNC | 32.6 (5.63) | 35.2 (4.43) | 27.5 (3.63) | 35.1 (3.29) | 23.9 (1.77) | 17.1 (0.39) |
|  | MLP | **91.3 (3.36)** | **83.5 (9.00)** | **88.6 (2.85)** | **65.5 (1.51)** | **60.6 (3.04)** | **83.4 (0.89)** |
|  | NK | 19.8 (0.39) | 17.8 (10.82) | 23.2 (0.41) | 9.5 (1.64) | 3.3 (7.11) | 6.0 (0.14) |

Table 1: Central tendency across five runs for our proposed metrics, SNC and NK, along with the chance-adjusted accuracy of an MLP on all neurons (MLP). The best in each block is in bold.

measuring the loss in function. We test whether $z_{\neq i}$ contains any information about $g_i$, by training an MLP to predict $g_i$ from $z_{\neq i}$. If the representation is disentangled, then this accuracy should be low, in comparison to an MLP that uses all neurons. Our second metric is $NK_i = Acc_z - Acc_{z_{\neq i}}$, where $A_x$ denotes the accuracy of an MLP trained on neurons $x$ to predict $g_i$. This is crucially different from most existing methods, which only measure the feature importance of each neuron individually, and so suffer the problems articulated in Section 3.2. Some existing works have used a similar idea to NK, (Sha and Lukasiewicz, 2021; Mathieu et al., 2016). These works partition the neurons, a priori, into two subsets, $A$ and $B$, representing distinct factors $a$ and $b$, respectively. Performance is then measured by training an MLP to predict $b$ from $A$ and $a$ from $B$. This technique, however, is only applicable where the set of neurons has been partitioned during training by the use of labels, whereas ours is more broadly applicable because we include a method for identifying which neurons to knock out. Secondly, we measure the difference with an MLP trained on all neurons, and so can distinguish between a disentangled representation and one that simply contains no information about the input. For prior works, the latter would score highly, but for NK, it would not, as then $Acc_z \approx Acc_{z_{\neq i}} \approx 0$.

## 5 EXPERIMENTAL EVALUATION

**Datasets.** We test our metrics and task on three datasets. **Dsprites** contains 737,280 black-and-white images with features $(x, y)$-coordinates, size, orientation and shape. **3dshapes** contains 480,000 images with features object/ground/wall colour, size, camera azimuth, and shape. **MPI3D** contains 103,680 images of objects at the end of a robot arm with features object colour, size and shape, camera height and azimuth, and altitude of the robot arm. All datasets have images of size $64 \times 64$.

**Implementation Details.** We test several popular DE models, $\beta$-VAE, FactorVAE and $\beta$-TCVAE, which are unsupervised, WeakDE, which is semisupervised, and PartedVAE, which can be trained unsupervised or semisupervised, and for which we test both settings. The details of these models are given in Section 2. Parted-VAE and Weak-DE are trained using the authors' public code for 100 epochs. Other models are trained using the library at https://github.com/YannDubs/disentangling-vae, all use default parameters. The MLPs and linear classification heads are trained using Adam, learning rate .001, $\beta_1$=0.9, $\beta_2$=0.999, for 75 epochs. The MLP has one hidden layer of size 256. MTD is trained using the author's code (obtained privately) with all default parameters, for 10 epochs. All experiments were performed on a single Tesla V100 GPU on an internal compute cluster.

### 5.1 SNC AND NK RESULTS

Table 1 shows the results of our two proposed metrics, SNC and NK on the three datasets described above. Each dataset includes a slightly different set of features, and displaying all features impairs readability, so we report the average across all features. Full results are given in the appendix.

The SNC accuracy is substantially lower than that of the full MLP. This suggests that each $g_i$ is represented more accurately by a distributed, non-linear entangled encoding across all neurons, rather than just by $z_i$. Although an MLP is a more powerful model, this should not help test set accuracy

unless there is relevant information in the input that it can leverage, i.e., distributed entanglements. The fact that MLP accuracy is much higher than SNC, suggests that such entanglements exist.

Distributed entanglements are also evidenced by the NK results. Here, there is often only a marginal drop in accuracy after removing the neuron that was supposed to contain all the relevant information, suggesting that much of the representation of $g_i$ is distributed over $z_{\neq i}$. Consider, e.g., WeakDE on Dsprites. SNC=5.8, meaning (roughly) that the neuron aligned to each factor predicts with 5.8% accuracy, so all non-aligned neurons predict that factor with accuracy <5.8%. Yet NK=5.9, meaning all these non-aligned neurons predict with 99.8-5.9 = 93.9% accuracy. (NK is the difference between the full MLP and the MLP with the aligned neuron knocked out.) This is much higher even than the sum of the accuracies for all of the seven non-aligned neurons, which would be $< 7 * 5.8 = 40.6\%$. This constitutes a distributed entanglement: the MI for a set is high, while the MI for each individual neuron is very low. Thus, Table 1 substantiates the argument Section 3.2 about the importance of distributed entanglements. showing that they are not merely a theoretical possibility, but that they also occur in practice.

## 5.2 DOWNSTREAM TASK: COMPOSITIONAL GENERALIZATION

Compositional generalization (CG) is the ability to combine familiar, learned concepts in novel ways. Here, we quantify CG using the same method as Xu et al. (2022). That is, we test whether the representations produced by a model can be used to correctly classify novel combinations of familiar features. The CG ability of machine learning models has mostly been studied in the context of language (Baroni, 2020). However, recently, a number of authors have observed the connection between CG and DE Zheng and Lapata (2021); Montero et al. (2020; 2022); Esmaeili et al. (2019); Zhang et al. (2022); Higgins et al. (2016). Disentangled models should be capable of performing CG, because they can represent each component separately and independently, whereas if there is entanglement between the different features, then the novel combination is out of distribution, and so the model will likely struggle to classify it correctly. Following Xu et al. (2022), we (1) randomly sample values for two features, e.g., shape and size, (2) form a test set of points with those two values for those two features, e.g., all points with size=0 and shape='square', and a train set of all other points, (3) train the VAE (or supervised model) on the train set, (4) encode both the train and test sets, (5) train and test an MLP to predict the generative factors from the encodings.

Table 2 shows results for the task of classifying novel combinations of familiar features. As well as the models from Section 5.1, we also report results for MTD, a fully supervised method (Sha and Lukasiewicz, 2021).

| | | Dsprites | | | 3dshapes | | | mpi3d | | |
| --- | --- | --- | --- | --- | --- | --- | --- | --- | --- | --- |
| | | shape | size | both | shape | size | both | shape | size | both |
| $\beta$-VAE | CG | 0.00 | 61.55 | 0.00 | 33.00 | 67.70 | 14.75 | 89.00 | 3.66 | 2.34 |
| | normal test set | 82.35 | 66.24 | **89.44** | 96.30 | 96.90 | **99.85** | 90.53 | 77.44 | **91.29** |
| $\beta$TCVAE | CG | 0.00 | 49.56 | 0.00 | 29.40 | 77.50 | 17.73 | 87.87 | 0.00 | 0.00 |
| | normal test set | 82.32 | 66.22 | **86.21** | 96.50 | 96.80 | **99.89** | 89.22 | 72.91 | **83.47** |
| FactorVAE | CG | 0.00 | 37.28 | 0.86 | 0.86 | 6.36 | 6.36 | 89.36 | 0.47 | 0.00 |
| | normal test set | 80.85 | 65.08 | **77.05** | 89.50 | 95.70 | **98.09** | 89.52 | 73.94 | **88.59** |
| PartedVAE | CG | 0.00 | 29.61 | 0.00 | 0.00 | 19.11 | 18.52 | 81.28 | 91.67 | 0.00 |
| | normal test set | 58.33 | 63.83 | **70.65** | 93.50 | 95.90 | **96.24** | 72.36 | 91.67 | **65.49** |
| PartedVAE-ss | CG | 0.00 | 31.83 | 0.00 | 0.00 | 30.33 | 30.33 | 72.30 | 0.00 | 0.00 |
| | normal test set | 39.47 | 38.48 | **48.48** | 76.70 | 94.20 | **88.93** | 71.20 | 25.70 | **60.61** |
| WeakDE | CG | 0.00 | 41.79 | 0.00 | 0.00 | 7.83 | 7.83 | 87.10 | 0.00 | 0.00 |
| | normal test set | 74.66 | 65.58 | **79.01** | 96.50 | 96.90 | **99.81** | 82.90 | 50.79 | **83.36** |
| MTD | CG | 0.00 | 0.03 | 0.00 | 0.00 | 0.50 | 0.00 | 23.30 | 0.00 | 0.00 |
| | normal test set | 99.66 | 100.00 | **99.56** | 100.00 | 100.00 | **100.00** | 99.80 | 95.01 | **95.36** |

Table 2: MLP classification accuracy for novel combinations of familiar features, denoted 'CG' and 'CG linear' respectively, and classification accuracy when the test set is chosen randomly, denoted 'normal test set'.

Due to the huge number of feature value combinations, it is not feasible to test all of them in enough detail to obtain reliable results. We restrict attention to shape and size, use five feature combinations for Dsprites, six for MPI3D and eight for 3dshapes (details in appendix), and report the average. The "normal test set" setting uses the same method except divides the train and test sets randomly. We adjust for chance agreement as $\max(0, (a - r)/(1 - r))$, where $a$ is the model accuracy, and $r$ is the chance agreement. Even restricting our attention to a single combination of feature types, our experimental results are already extensive involving $\sim 200$ VAE models, and $> 1000$ classification heads (see appendix). An empirical study of multiple feature types would require 10-100x more, which would be a valuable future contribution, but is outside the scope of the present work.

The accuracy for identifying novel combinations of familiar features is generally low, often at the level of random guessing (i.e., 0 after adjusting for chance agreement). This is even true for MTD, the supervised model. In the 'normal test set' setting, every model is capable of classifying the unseen data accurately, which shows that it is the novelty of the combination that is degrading performance.

There is perhaps a danger that the MLP itself entangles the two features. For example, if yellow circles are excluded then, when classifying shape, the MLP could learn that whenever the "colour" neuron indicates "yellow", it should place low probability mass on "circle". We feel this is unlikely to affect results significantly, as the relationship between the "size" neuron and the value of shape would be highly irregular and present only for a small subset of data points. To make sure of this, repeat the experiment using a linear classifier instead of an MLP, so this non-linear relationship could not be learnt. The performance of the linear classifier is as low or lower, almost never above a random baseline, which suggests the poor performance is not due to the MLP itself entangling the two factors.

There is a clear difference across datasets: results on Dsprites and MPI3D are essentially always at the level of random guessing, whereas those on 3dshapes are more promising, reaching nearly 30% (chance adjusted) for some models. PartedVae performs best, especially the semi-supervised variant, though perhaps surprisingly, the other recent semi-supervised method, WeakDE, performs less well.

Some prior works have claimed their model can meaningfully represent novel combinations: Higgins et al. (2016) display figures of reconstructed chairs with a round bottom for certain latent traversals and Esmaeili et al. (2019) present reconstructions for MNIST digits with certain combinations of digits and features (e.g., line thickness) excluded. Conflicting results were found by Montero et al. (2020; 2022), who claimed that ability to represent novel combinations was unrelated to the degree of DE. However, those prior works have mostly only examined the reconstructions by the decoder and so do not provide sufficient evidence to make claims about the internal representation. Being able to reconstruct an image accurately does not establish anything about the internal representation, it could just be the result of learning the identity function. The key question is not what the decoder reconstructs from the encoding or from a latent traversal or from a sampled latent vector, it is what representation the *encoder* produces. Also, the experiment by Montero et al. (2020) attempted to quantify CG performance as the decoder's pixel loss, which has long been observed to be a poor measure of the quality of a generated image Oprea et al. (2020); Higgins et al. (2016). Montero et al. (2022) does manage to assess the encoder, but this is only a qualitative assessment, so cannot determine quantitative correlation. On the other hand, classifying novel combinations, as we do, following Xu et al. (2022), is both quantitative, and is able to assess the encoder rather than the decoder. Unlike Higgins et al. (2016); Esmaeili et al. (2019), our experiments mostly show poor performance of existing DE models at CG. They also differ from Montero et al. (2020), as they reveal a correlation between DE and CG, with almost all metrics, and most strongly with our metrics.

### 5.3 OUR METRICS PREDICT COMPOSITIONAL GENERALIZATION

Table 3 shows the Pearson correlation of our metrics and existing metrics, computed from the disentanglement_lib library, with CG performance. We restrict our metrics to just size and shape, because they are the relevant features for this task. We also tried restricting existing metrics to these two features only, without much change in results; see appendix. We show correlation on 3dshapes, and across all datasets. There is no insight to be gained from the correlation on Dsprites only or MPI3D only, since performance there is rarely above random, so all correlations will be essentially zero. Although performance is generally low on CG, there is still enough variation to measure a meaningful correlation. Some models (PVAE on 3dshapes) achieve up to 30% even after chance adjusting, which is significantly higher than the random guessing of the lower-scoring models. A good DE metric should distinguish between models at the higher and lower ends of this range.

|  | SNC | NK | MIG | SAP | IRS | D | C | I | DCI | MED |
|---|---|---|---|---|---|---|---|---|---|---|
| 3dshapes | **.849** | .710 | .800 | .726 | .668 | .472 | .819 | .740 | .717 | .785 |
| all datasets | **.850** | .716 | .571 | .311 | .106 | .457 | .414 | .535 | .493 | .566 |

Table 3: Correlation of DE metrics with accuracy on novel combinations. Best in bold, second best underlined.

The first observation is that all metrics are at least weakly correlated with compositional generalization performance, with some showing a moderate to strong correlation. This contradicts Montero et al. (2020), who claimed to find no relationship between DE and CG. As argued in Section 5.2, our method for measuring CG performance is more indicative of the encoding quality. Additionally, we test six models, eight metrics and three datasets, whereas Montero et al. (2020) test only two models, one metric and two datasets. Later, the same authors conduct a more thorough investigation of CG in DE models Montero et al. (2022). They show that the encoder often maps CG test examples to the wrong region of latent space, despite achieving high DCI score, and so conclude that DE does not offer much benefit to CG performance. However, the discrepancy between DCI and CG can also be explained by DCI overestimating DE, as in Theorem 3.1, and experimentally in Cao et al. (2022). The conclusion that our work supports is that DE and CG are indeed closely related, but that this relationship is obscured by the fact that, so far, DE has been overestimated by all existing metrics, including DCI. That is, both DE and CG are low for all models, but they are correlated.

The consistent correlation between DE and CG evidenced by Table 3 also shows that there is more than enough variance in the CG scores to obtain a meaningful correlation. If all models were essentially at random guessing, correlation with any metric would be due only to chance, and we would expect correlations close to zero, but this is not the case. There is clearly enough variation in the CG results to identify a meaningful relationship to metric score, because the correlation is strong, far above zero, for most metrics, and far above statistical significance ($p < 0.01$) for our metrics (calculation in appendix). This further validates the suitability of CG as a downstream task to evaluate DE metrics.

SNC correlates more strongly than NK. This is expected, as SNC measures the extent to which information about the generative factors is encoded in a disentangled way, whereas NK penalizes representations that also encode information in a redundant, entangled way. NK is perhaps more relevant than SNC to interpretability, where we want to know that a given factor is encoded by a single neuron *only*, whereas SNC is more relevant for downstream performance. A similar comparison can be made between the components of DCI: C and I measure the extent to which the information is present in a disentangled way Thus, they are loosely analogous, in the context of DCI, to our SNC, and, like SNC, are a better predictor of downstream performance than D.

# 6 DISCUSSION

**Limitations and Future Work.** One limitation of our work is the focus on only two features in combination. Our experimental results are already extensive involving $\sim 200$ VAE models, and $>$ 1000 classification heads, but an interesting future work would be to perform the same analysis for a different feature combination. This could investigate, for example, whether novel combinations of simple features like $(x, y)$ coordinates, are easier to recognize than novel size-shape combinations. Another limitation is the applicability of our metrics only to strong DE. As we have argued, strong DE and weak DE are fundamentally different objectives, and so we claim that they should be assessed by different metrics. This implies that it is not possible to design a metric that works for both forms. Instead, one could consider a modification of our metrics that work in the case of weak DE.

**Conclusion.** In this paper, we identified two common flaws in existing disentanglement metrics: incorrect alignment of generative factors with neurons and conflating the importance of individual neurons with that of sets of neurons, and showed how these problems occur in real-world examples. We then introduced two new metrics, single-neuron classification and neuron knockout, which avoid these problems. Next, we proposed the classification of novel combinations of familiar features as a real-world downstream task against which to compare disentanglement metrics, and showed that our metrics are strongly predictive of performance on this task, more strongly than existing metrics.

ETHICS STATEMENT

We have considered the potential ethical implications of this work, and do not believe there are any concerns in this respect. The data used is all publicly available and impersonal, there is no use of human subjects, and there are no obvious malicious applications of our work. There are also no legal issues or issues are conflict with sponsorship.

REPRODUCIBILITY STATEMENT

In Section 4, we describe how our metrics are calculated. In Section 5, we describe the structure of the experiments we ran, and give implementation details and reference to publicly available code. Most importantly, we release all the code to reproduce our experiments in an anonymous repo https://github.com/anon296/anon.

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

## A    Calculations from the Examples in Figure 1

Here, we show the calculation of MIG, DCI, and SAP, as well as our metrics, from the example of Table 1 in Section 3. We are interested in the change of these metrics when $z_1$ The others all, incorrectly, decrease, while ours, correctly, increase.

**MIG.**    Let $g_1$ denote shape and $g_2$ denote colour, and let $H$ denote entropy and $MI$ denote mutual information. We then calculate the MIG for the model in Table 1 as follows. First, note that

$$MI(z_2, g_1) = MI(z_2, g_2) = 0\,, \tag{2}$$

because $z_2$ is just noise. Then, assuming balanced classes, we have $H(g_1) = H(g_2) = 1$. As $z_1$ encodes each to an accuracy of 75%, the conditional entropy is

$$-0.75 \log(0.75) - (0.25) \log 0.25 = 0.8113\,,$$

and so

$$MI(z_1, g_1) = MI(z_1, g_2) = 1 - 0.8113 = 0.1887\,.$$

The MIG, which is identical for both features, is then

$$MI(z_1, g_1) - MI(z_2, g_1) = MI(z_1, g_2) - MI(z_2, g_2) =$$
$$0.1887 - 0 = 0.1887\,.$$

Now, we calculate MIG for the second described model. Here,

$$MI(z_1, g_1) = MI(z_1, g_2) = 0.1887\,,$$

as above, and also $MI(z_2, g_1) = 0$ as above. Now, however,

$$MI(z_2, g_2) = 1 - (-0.7 \log(0.7) - (0.3) \log 0.3) =$$
$$1 - 0.8813 = 0.1187\,,$$

So, the MIG for $g_1$ is the same as for the first model, but the MIG for $g_2$ is

$$MI(z_1, g_2) - MI(z_2, g_2) = 0.1887 - 0.1187 = 0.07\,.$$

**DCI.**    The $I$ component is unchanged at 75%. To calculate $D$ and $C$, we need some measure of the importance or strength of the connection of each neuron to each feature. DCI usually measures feature importance with a classifier, but of course we cannot train a classifier on a theoretical example. The only information is the accuracy to which each neuron encodes each feature, and this provides a reasonable way to quantify feature importance for this example. There are three obvious ways in which one could use the given accuracies in our example to quantify feature importance: we could use the accuracy values themselves, we could use the normalized accuracy values so that the random noise neuron is measured as being of zero importance, and we could use the mutual information scores (as used for MIG). The follow code excerpt computes $D$ and $C$ from the disentanglement_lib implementation for each of these three choices. In all three, the average decreases, when it should decrease.

```
from dci import disentanglement, completeness
import numpy as np

def print_befores_and_afters(b,a):
    bd = disentanglement(b)['avg']
    ad = disentanglement(a)['avg']
    bc = completeness(b)['avg']
    ac = completeness(a)['avg']
    bavg = (bd+bc)/2
    aavg = (ad+ac)/2
    print(f'Before:\n{bd}, {bc}, {bavg}')
    print(f'After:\n{ad}, {ac}, {aavg}')

print('USING ACC AS IMPORTANCE')
```

```
b = np.array([[.75,.75],[.5,.5]])
a = np.array([[.75,.75],[.5,.7]])
print_befores_and_afters(b,a)

print('USING NORMED ACC AS IMPORTANCE')
b = np.array([[.5,.5],[.0,.0]])
a = np.array([[.5,.5],[.0,.2]])
print_befores_and_afters(b,a)

print('USING MUTUAL INFO AS IMPORTANCE')
b = np.array([[.1887,.1887],[.0,.0]])
a = np.array([[.1887,.1887],[.0,.1187]])
print_befores_and_afters(b,a)
```

**SAP** If we take the accuracy as roughly equal to the $R^2$ coefficient, then the SAP score for both factors before the change is 0.25, whereas after the change it is 0.25 for colour and 0.05 for shape, so the average decreases to 0.15.

**SNC and NK.** Our SNC metric is the average of the two chance-adjusted accuracies, $(0.5 + 0)/2 = 0.25$. NK is the drop in chance-adjusted accuracy after removing the aligned neuron, which is equal to 0.25 for colour and 0 for shape. In the variant where $z_1$ encodes shape to an accuracy of 70%, SNC becomes $(0.5 + 0.4)/2 = 0.45$, so correctly increases. NK is unchanged.

## B  HINTON DIAGRAMS FOR MISALIGNED FACTORS

Existing metrics simply assign each generative factor to the neuron that is most informative about it, as measured by mutual information or the weight in a linear classifier. Section 3 showed an example of this producing an incorrect alignment where multiple different factors are assigned to the same neuron. Our method, in contrast, enforces that all assignments are unique. Here we show some further examples of misalignments resulting from the method used by existing metrics.

The following Hinton diagrams show the size of the square at $(i, j)$ is proportional to the mutual information between factor $j$ and neuron $i$.

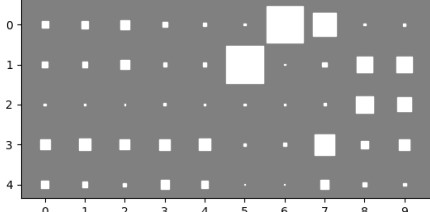

Figure 2: Hinton diagram showing alignment of factors (y-axis) to neurons (x-axis) for $\beta$-VAE on Dsprites. Existing metrics incorrectly align both factor 3 and factor 4 to neuron 7, our method correctly aligns factor 3 to neuron 7 and factor 4 to neuron 3.

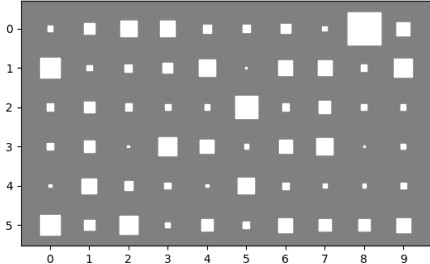

Figure 3: Hinton diagram showing alignment of factors (y-axis) to neurons (x-axis) for $\beta$-VAE on 3dshapes. Existing metrics incorrectly align both factor 2 and factor 4 to neuron 5, our method correctly aligns factor 2 to neuron 5 and factor 4 to neuron 1.

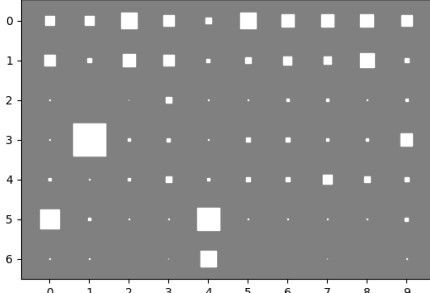

Figure 4: Hinton diagram showing alignment of factors (y-axis) to neurons (x-axis) for $\beta$-VAE on MPI3D. Existing metrics incorrectly align both factor 5 and factor 6 to neuron 4, our method correctly aligns factor 5 to neuron 0 and factor 6 to neuron 4.

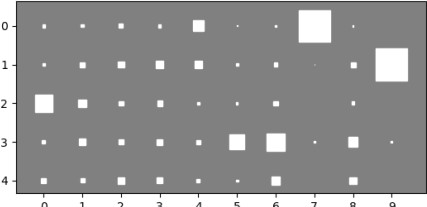

Figure 5: Hinton diagram showing alignment of factors (y-axis) to neurons (x-axis) for $\beta$-TCVAE on Dsprites. Existing metrics incorrectly align both factor 3 and factor 4 to neuron 6, our method correctly aligns factor 3 to neuron 6 and factor 4 to neuron 8.

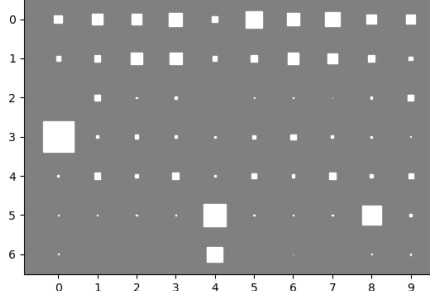

Figure 6: Hinton diagram showing alignment of factors (y-axis) to neurons (x-axis) for $\beta$-TCVAE on MPI3D. Existing metrics incorrectly align both factor 1 and factor 4 to neuron 3, our method correctly aligns factor 1 to neuron 3 and factor 4 to neuron 7.

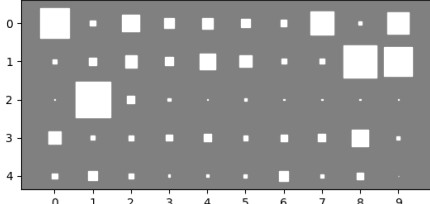

Figure 7: Hinton diagram showing alignment of factors (y-axis) to neurons (x-axis) for FactorVAE on Dsprites. Existing metrics incorrectly align both factor 1 and factor 3 to neuron 8, our method correctly aligns factor 1 to neuron 8 and factor 3 to neuron 7.

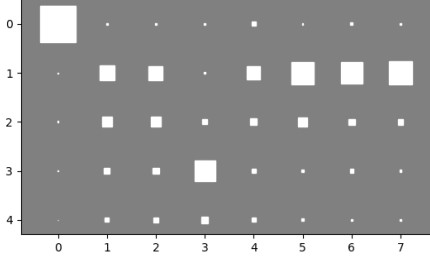

Figure 8: Hinton diagram showing alignment of factors (y-axis) to neurons (x-axis) for PartedVAE on Dsprites. Existing metrics incorrectly align both factor 3 and factor 4 to neuron 3, our method correctly aligns factor 3 to neuron 3 and factor 4 to neuron 2.

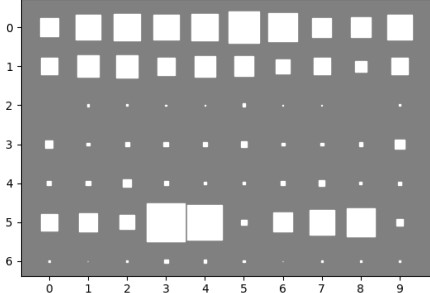

Figure 9: Hinton diagram showing alignment of factors (y-axis) to neurons (x-axis) for WeakDE on MPI3D. Existing metrics incorrectly align both factor 0 and factor 2 to neuron 5, our method correctly aligns factor 0 to neuron 5 and factor 2 to neuron 1.

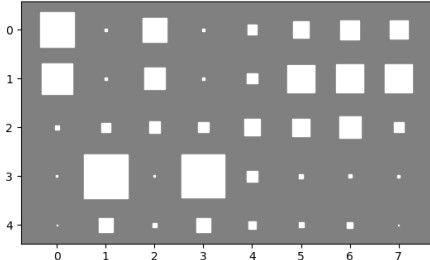

Figure 10: Hinton diagram showing alignment of factors (y-axis) to neurons (x-axis) for PartedVAE-semisupervised on Dsprites. Existing metrics incorrectly align both factor 0 and factor 1 to neuron 0, our method correctly aligns factor 0 to neuron 0 and factor 1 to neuron 7.

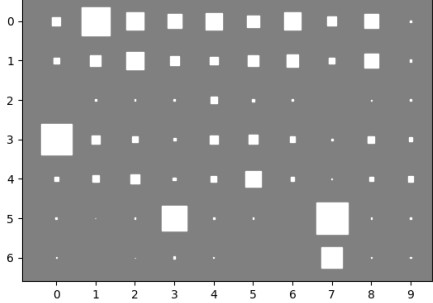

Figure 11: Hinton diagram showing alignment of factors (y-axis) to neurons (x-axis) for FactorVAE on MPI3D. Existing metrics incorrectly align both factor 5 and factor 6 to neuron 7, our method correctly aligns factor 5 to neuron 3 and factor 6 to neuron 7.

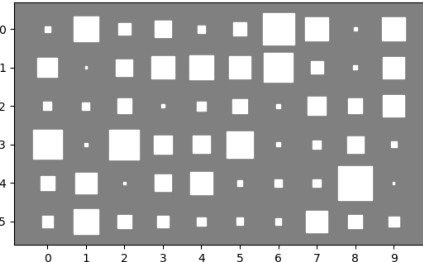

Figure 12: Hinton diagram showing alignment of factors (y-axis) to neurons (x-axis) for WeakDE on 3dshapes. Existing metrics incorrectly align both factor 0 and factor 1 to neuron 6, our method correctly aligns factor 0 to neuron 6 and factor 1 to neuron 4.

|  |  | $\beta$-VAE | $\beta$-TCVAE | FactorVAE | PartedVAE | PartedVAE-ss | weakde |
|---|---|---|---|---|---|---|---|
| Dsprites | SNC | 24.8 (3.20) | 41.3 (3.00) | 15.1 (1.78) | 16.1 (4.54) | 19.0 (4.80) | 5.8 (0.75) |
|  | linear | 46.2 (2.54) | 49.5 (1.60) | 30.8 (2.43) | 36.0 (2.58) | 26.3 (10.03) | 28.6 (2.00) |
|  | MLP | **89.4 (3.95)** | **86.2 (1.10)** | **77.1 (4.10)** | **70.7 (10.33)** | **48.5 (9.08)** | **99.8 (3.50)** |
|  | NK | 39.9 (1.11) | 31.3 (4.10) | 32.9 (1.45) | 34.9 (4.23) | 14.5 (8.17) | 5.9 (1.17) |
| 3dshapes | SNC | 19.9 (4.70) | 20.1 (2.06) | 16.9 (3.36) | 43.8 (24.97) | 68.3 (9.55) | 15.4 (2.11) |
|  | linear | 79.0 (1.60) | 76.3 (3.47) | 65.2 (4.74) | 70.8 (6.73) | 65.4 (11.40) | 73.3 (18.17) |
|  | MLP | **99.8 (3.50)** | **99.9 (2.50)** | **98.1 (7.24)** | **96.2 (6.83)** | **88.9 (14.53)** | **99.8 (0.68)** |
|  | NK | 8.0 (0.20) | 8.1 (0.11) | 16.8 (2.94) | 60.2 (6.19) | 41.8 (8.07) | 2.5 (0.18) |
| MPI3D | SNC | 32.6 (5.63) | 35.2 (4.43) | 27.5 (3.63) | 35.1 (3.29) | 23.9 (1.77) | 17.1 (0.39) |
|  | linear | 51.3 (2.73) | 52.1 (4.20) | 46.9 (2.52) | 49.4 (2.32) | 43.0 (5.11) | 46.2 (1.57) |
|  | MLP | **91.3 (3.36)** | **83.5 (9.00)** | **88.6 (2.85)** | **65.5 (1.51)** | **60.6 (3.04)** | **83.4 (0.89)** |
|  | NK | 19.8 (0.39) | 17.8 (10.82) | 23.2 (0.41) | 9.5 (1.64) | 3.3 (7.11) | 6.0 (0.14) |

Table 4: Central tendency across five runs for our proposed metrics, SNC and NK, along with the accuracy predicting each factor using an MLP on all neurons (MLP) and a linear classifier on all neurons (linear). The best accuracy in each block is in bold.

## C RESULTS WITH LINEAR CLASSIFIERS

Table 4 shows the results from 1 alongside the accuracy from a linear classifier, which aligns more closely with the conception of disentanglement from Ridgeway and Mozer (2018).

# D  PROOF OF THEOREM 1

**Theorem D.1.** *As the number of neurons and factors in the representation increases, the $D$ and $C$ components of DCI can, under a very broad class of feature importance measures, including gradient-boosted trees, assign a score that is arbitrarily close to perfect, even though the model is completely entangled in the sense that no neuron, by itself, contains any information about any generative factor.*

*Proof.* Let $z_0, \ldots z_{m-1}$ be a set of neurons encoding factors $g_0, \ldots, g_{n-1}$, with $m \geq n$. Let $k_0, \ldots, k_{n-1}$ denote the number of different values of each feature, that is, $g_i$ can take on $k_i$ different values, for each $i$. Assume WLOG that each neuron is normalized to the interval $[0, 1]$. This is equivalent to any unnormalized representation via a simple scaling of weights in the output of the encoder.

Let $b(z, k)$ denote the value of binning $z$ into $k$ different bins:

$$b(z, k) = \lfloor (zk) \rfloor,$$

and let $c(z, k)$ denote the scaled remaining portion of $z$ that does not contribute to its binned value:

$$c(z, k) = kz - b(z, k).$$

Then, suppose that the representation function is as follows:

$$g_i = b(z_i, k_i) + b(c(z_j, k_j), k_i) \mod k_i, \tag{3}$$

where $j = i + 1 \mod n$. Intuitively, this means that we divide the significant bits of $z_i$ into two portions, the first chunk of significant bits is used to compute $b(z_i, k_i)$ and contributes towards $g_i$, the remaining chunk is used to compute $c(z_i, k_i)$ and contributes towards $g_{i-1}$ (where $i - 1$ is taken with modulo arithmetic). For any values of $g_0, \ldots, g_{n-1}$, such a representation exists, by setting

$$b(z_i, k_i) = g_i - b(c(z_j, k_j) \mod k_i$$
$$c(z_j, k_j) = g_i - b(c(z_j, k_j) \mod k_i.$$

and then computing a solution for the system of $2n$ linear equations $2n$ unknowns. Together $b(z_i, k_i)$ and $c(z_i, k_i)$, along with $k_i$, uniquely determined $z_i$.

Clearly, $\forall l \neq i, j, I(z_l; g_i) = 0$, where $I$ denotes mutual information, because equation 3 does not involve $z_l$ at all. We now show that, when $g_i$ is uniformly distributed, $I(z_i; g_i) = I(z_j; g_i) = 0$ as well, establishing that the representation is completely entangled in the sense that $I(z; g) =$ for all neurons $z$ and factors $g$.

*Remark D.2.* There is currently interest in applying disentanglement to data where the values of each feature are non-uniformly distributed, and we expect the following proof could be extended to cover such cases. However, the present form is enough to show the flaw in DCI. As well as the non-uniform case, a good metric should also of course be able to give the correct answer in the uniform case. In other words, giving the correct answer in the uniform case is a necessary but not sufficient condition for a good metric. We show that DCI fails in the uniform case, and so fails to be a good metric.

If $g_i$ is uniformly distributed on $\{0, \ldots, k_i - 1\}$, then one solution to the system of equations is where $b(z_i, k_i)$ and $b(c(z_j, k_j), k_i)$ are also uniformly distributed. This follows from the fact that, on any countable discrete group $G$, the convolution of two uniform probability measures is again uniform, and the fact that addition of two random variables, as in equation 3, corresponds to a convolution, in this case over the group of integers modulo $k_i$. Thus, setting $p(b(z_i, k_i) = b(c(z_j, k_j), k_i) = \mathcal{U}(k_i)$, gives, $p(g_i) = \mathcal{U}(k_i)$, where $\mathcal{U}(k)$ is the discrete uniform distribution on $k$ elements. This gives the following:

$$p(g_i | z_i = z) = p(b(z_i, k_i) = g_i - b(k_j c(z, k_j), k_i)) = \mathcal{U}(k) \, \forall z \in [0, 1]$$
$$p(g_i | z_j = z) = p(b(z_i, k_i) = g_i - b(k_j c(z, k_j), k_i)) = \mathcal{U}(k) \, \forall z \in [0, 1],$$

where the second equality in each line follows from the fact that $b(k_j c(z, k_j), k_i)$ is a constant that doesn't depend on $g_i$, and that subtraction of a constant constitutes a bijection in the group of

integers modulo $k_i$. Therefore, the conditional distribution of $g_i$ given $z_i$ or $z_j$ is equal to the marginal distribution of $g_i$. In particular, the entropies are equal, giving zero mutual information:

$$I(g_i; z_i) = H(g_i) - H(g_i|z_i) = \mathcal{U}(k_i) - \mathcal{U}(k_i) = 0$$
$$I(g_i; z_j) = H(g_i) - H(g_i|z_j) = \mathcal{U}(k_i) - \mathcal{U}(k_i) = 0 \,.$$

This completes the first part of the proof. Now we must show that DCI assigns a high score to such a representation. In principle, DCI can use any measure of feature importance, however the following argument makes a very general assumption about the feature importance measure, which includes all but pathological feature importance measures, and so covers all cases of practical interest. Let $R$ be the $m \times n$ matrix of feature importance, where $R_{ji}$ is the importance of neuron $j$ in predicting feature $i$. The assumption is that, if $g_i$ is uniquely defined by an equation that does not involve $z_j$, and $z_j$ gives no indication as to the value of $g_i$, then $R_{ji} = 0$. We can formalize "gives no indication as to the value of" as $I(z_j, g_i) = 0$ and, for any non-empty set of neurons $\bar{z}$, $I(g_i; z_j, \bar{z}) = I(g_i; \bar{z})$. This means

$$R = \begin{pmatrix} R_{1,1} & 0 & 0 & \dots & 0 & 0 & R_{1,n-1} \\ R_{2,1} & R_{2,2} & 0 & \dots & 0 & 0 & 0 \\ 0 & R_{3,2} & R_{3,3} & \dots & 0 & 0 & 0 \\ 0 & 0 & 0 & \dots & R_{n-2,n-3} & R_{n-2,n-2} & 0 \\ 0 & 0 & 0 & \dots & 0 & R_{n-1,n-2} & R_{n-1,n-1} \end{pmatrix} \,.$$

DCI normalizes feature importance across all features, $P_{j,i} = R_{j,i} / \sum_{k=0}^{n-1} R_{j,k}$. Assuming the feature importances are not all zero for any feature, in which case DCI is undefined because of zero-division, both $D$ and $C$ are equal to one minus the entropy of some distribution over $n$ elements, with either exactly one or exactly two elements given non-zero probability. The entropy of such a distribution is maximized when both non-zero elements are equal to $\frac{1}{2}$, giving entropy

$$-\sum_{i=0}^{n} p(x) \log p(x) = 2 \log_N 2 = \frac{2}{\log n} \,.$$

Thus, $D, C \geq 1 - 2/\log n$ and so

$$\lim_{n \to \infty} D = \lim_{n \to \infty} C = 1 \,.$$

This holds for all features individually, and so by symmetry, it holds on the weighted sum across all features. $\qquad\square$

*Remark* D.3. We do not need very many neurons before the lower bound on both D and C becomes significant. For only 16 neurons, $D, C \geq 0.5$, for 64 neurons, $D, C \geq 0.67$.

# E CORRELATIONS OF DIFFERENT VERSIONS OF EXISTING METRICS WITH COMPOSITIONAL GENERALIZATION PERFORMANCE

As reported in Section 5.3, our metrics, restricted to the two novel feature types of size and shape, are more predictive of performance on compositional generalization than are existing metrics. When calculating the scores for other metrics, we took the average across all features. Table E shows the correlation when restricting to just the features of shape and size. We show both the mean of these two features and the product (as for our metrics) of these two features. Note that the D component from DCI, and the IRS metric, are not computed feature-wise, so we cannot restrict it to just two features. These alternative variants of existing metrics perform better in some cases and worse in others. Overall, they are about equally predictive and, importantly, still all less predictive than our metrics, especially the SNC metric.

|                  | all datasets | 3dshapes  |
| ---------------- | ------------ | --------- |
| SNC              | **0.850**    | **0.849** |
| NK               | *0.716*      | 0.710     |
| MIG              | 0.571        | 0.800     |
| MIG product of 2 | 0.570        | 0.640     |
| MIG mean of 2    | 0.514        | 0.786     |
| SAP              | 0.311        | 0.726     |
| SAP product of 2 | 0.626        | 0.544     |
| SAP mean of 2    | 0.453        | 0.625     |
| IRS              | 0.106        | 0.668     |
| D                | 0.457        | 0.472     |
| C                | 0.414        | *0.819*   |
| C product of 2   | 0.288        | 0.681     |
| C mean of 2      | 0.280        | 0.770     |
| I                | 0.535        | 0.740     |
| I product of 2   | -0.052       | 0.605     |
| I mean of 2      | 0.119        | 0.682     |
| DCI              | 0.493        | 0.717     |

Table 5: Correlation (Pearson) of our metrics, and existing metrics, with accuracy on novel combinations. Best results in bold, second best italicized.

## F  STATISTICAL SIGNIFICANCE CALCULATION

Here, we show the statistical significance of the correlations from Table 3. For the 'all datasets" experiment, there are 6 methods on 3 datasets, so 18 data points, giving t-value

$$\frac{0.85\sqrt{18-2}}{\sqrt{1-0.85^2}} \approx 6.45 \,,$$

which gives a p-value $< $ 1e-5.

For the '3dshapes' experiment, there are 6 methods on 1 datasets, so 6 data points, giving t-value

$$\frac{0.85\sqrt{6-2}}{\sqrt{1-0.85^2}} \approx 3.23 \,,$$

which gives a p-value $< 0.033$, so still significant at $p < 0.05$.

# G  RESULTS OF ALL METRICS

Table 6 shows the results of our metrics and existing metrics on the datasets and methods we test on, averaged over five runs for Dsprites, eight for 3dshapes and six for MPI3D.

| | Dsprites | | | | | | 3dshapes | | | | | | mpi3d | | | | | |
| | betaH | btcvae | factor | pvae | pvae-ss | weakde | betaH | btcvae | factor | pvae | pvae-ss | weakde | betaH | btcvae | factor | pvae | pvae-ss | weakde |
|---|---|---|---|---|---|---|---|---|---|---|---|---|---|---|---|---|---|---|
| SNC | 6.954 | 8.302 | 6.400 | 3.310 | 4.339 | 0.678 | 8.354 | 7.494 | 5.747 | 12.369 | 23.674 | 10.328 | 2.830 | 2.718 | 2.573 | 6.054 | 0.990 | 0.143 |
| NK | 0.072 | 0.116 | 1.567 | 3.015 | 0.000 | 0.002 | 0.028 | 0.000 | 0.166 | 16.992 | 13.608 | 0.033 | 0.341 | 0.351 | 0.536 | 0.075 | 0.001 | 0.062 |
| MIG2p | 0.012 | 0.036 | 0.008 | 0.004 | 0.001 | 0.000 | 0.005 | 0.004 | 0.000 | 0.062 | 0.037 | 0.001 | 0.001 | 0.001 | 0.000 | 0.000 | 0.001 | 0.000 |
| MIG | 0.085 | 0.144 | 0.088 | 0.070 | 0.058 | 0.014 | 0.063 | 0.062 | 0.032 | 0.279 | 0.255 | 0.026 | 0.104 | 0.171 | 0.086 | 0.086 | 0.023 | 0.013 |
| MIG2m | 0.130 | 0.204 | 0.177 | 0.068 | 0.047 | 0.014 | 0.069 | 0.051 | 0.018 | 0.265 | 0.236 | 0.038 | 0.023 | 0.030 | 0.012 | 0.072 | 0.046 | 0.005 |
| SAP | 0.052 | 0.048 | 0.062 | 0.034 | 0.030 | 0.012 | 0.043 | 0.033 | 0.030 | 0.219 | 0.176 | 0.019 | 0.099 | 0.122 | 0.183 | 0.197 | 0.094 | 0.012 |
| SAP2p | 0.006 | 0.004 | 0.009 | 0.004 | 0.003 | 0.001 | 0.003 | 0.000 | 0.002 | 0.070 | 0.034 | 0.001 | 0.001 | 0.001 | 0.002 | 0.000 | 0.000 | 0.000 |
| SAP2m | 0.106 | 0.098 | 0.140 | 0.054 | 0.050 | 0.024 | 0.056 | 0.020 | 0.047 | 0.200 | 0.151 | 0.038 | 0.025 | 0.031 | 0.038 | 0.040 | 0.090 | 0.012 |
| IRS | 0.448 | 0.558 | 0.558 | 0.610 | 0.732 | 0.491 | 0.385 | 0.490 | 0.510 | 0.720 | 0.708 | 0.466 | 0.427 | 0.604 | 0.545 | 0.739 | 0.622 | 0.531 |
| C2p | 0.349 | 0.472 | 0.521 | 0.264 | 0.739 | 0.047 | 0.232 | 0.455 | 0.141 | 0.703 | 0.448 | 0.131 | 0.279 | 0.341 | 0.326 | 0.199 | 0.252 | 0.040 |
| C2m | 0.576 | 0.686 | 0.750 | 0.505 | 0.395 | 0.219 | 0.494 | 0.505 | 0.387 | 0.842 | 0.714 | 0.369 | 0.554 | 0.551 | 0.545 | 0.859 | 0.503 | 0.230 |
| I2p | 0.544 | 0.644 | 0.533 | 0.319 | 0.739 | 0.175 | 0.149 | 0.419 | 0.114 | 0.515 | 0.322 | 0.155 | 0.102 | 0.265 | 0.213 | 0.157 | 0.539 | 0.164 |
| I2m | 0.745 | 0.783 | 0.733 | 0.580 | 0.503 | 0.424 | 0.387 | 0.395 | 0.365 | 0.826 | 0.698 | 0.407 | 0.327 | 0.372 | 0.305 | 0.508 | 0.840 | 0.442 |
| D | 0.574 | 0.721 | 0.483 | 0.474 | 0.400 | 0.189 | 0.415 | 0.454 | 0.488 | 0.841 | 0.769 | 0.335 | 0.515 | 0.546 | 0.473 | 0.472 | 0.374 | 0.257 |
| C | 0.544 | 0.595 | 0.453 | 0.461 | 0.459 | 0.084 | 0.320 | 0.372 | 0.296 | 0.792 | 0.687 | 0.219 | 0.457 | 0.477 | 0.437 | 0.331 | 0.231 | 0.134 |
| I | 0.591 | 0.629 | 0.481 | 0.492 | 0.491 | 0.095 | 0.342 | 0.394 | 0.308 | 0.801 | 0.720 | 0.228 | 0.476 | 0.483 | 0.439 | 0.339 | 0.243 | 0.139 |
| DCI | 0.012 | 0.040 | 0.009 | 0.005 | 0.000 | 0.000 | 0.003 | 0.001 | 0.000 | 0.075 | 0.039 | 0.001 | 0.001 | 0.001 | 0.000 | 0.000 | 0.002 | 0.000 |
| MED | 0.068 | 0.113 | 0.071 | 0.050 | 0.042 | 0.008 | 0.048 | 0.044 | 0.029 | 0.231 | 0.213 | 0.026 | 0.071 | 0.147 | 0.076 | 0.079 | 0.020 | 0.013 |

Table 6: Results of existing metrics on the datasets and methods we test on. The suffix '2p' and '2m' indicate, respectively, the product and mean across the two features being compositionally generalized, size and shape. When this suffix is absent, the figure is the mean across all features. Our own metrics, SNC and NK, are shown as the product across size and shape, because those are the figures used to calculate the correlation as reported in the main paper.

# H  FULL RESULTS

Each of the following tables shows all results for a particular dataset and method combination. That is, each table shows results for single-neuron classification (SNC), neuron knockout (NK1 and NK2) and recognition of novel combinations of familiar features (NCFF) under the various settings described in the main paper.

For the compositional generalization settings, we indicate the values of two features excluded. Recall that we always exclude a combination of size and shape. So, for example "NC 3-2" means that the disentanglement model and the classifer trained on top of it, used a train set that excluded exactly those data points with size 3 and shape 2 (under some arbitrary ordering of the values of shape, e.g. 0=square, 1=circle, 2=crescent).

For all settings, we report the results for all features (where measured). The feature lists for each data set are as follows:

- **Dsprites:** x-position (x), y-position (y), object size (size), object orientation (orient), and object shape (shape)
- **3dshapes:** floor colour (floor h), wall colour (wall h), object size (size), camera azimuth (orient), object shape (shape), and object colour (object h)
- **MPI3D:** azimuth of robot arm (hor), altitude of robot arm (vert), size of object (size), colour of object (obj h), shape of object (shape), height of camera above the object (cam he), and background colour (bg h)

We also report the accuracy on both of the novel features, i.e. the fraction of the points for which the classifier correctly predicted both size and shape. This is shown in the "NC" column.

|  |  | x | y | size | orient | shape | NC |
|---|---|---|---|---|---|---|---|
| Normal Test set 0 | SNC | 40.78 | 46.84 | 44.80 | 7.34 | 52.17 | 23.17 |
|  | linear | 70.11 | 72.09 | 48.02 | 7.24 | 44.31 | 17.55 |
|  | NK1 | 17.59 | 20.80 | 48.83 | 50.04 | 98.87 | - |
|  | MLP | 89.63 | 92.87 | 99.04 | 65.54 | 99.64 | 98.83 |
| Normal Test set 1 | SNC | 29.74 | 32.44 | 45.90 | 7.17 | 49.99 | 23.52 |
|  | linear | 57.68 | 62.44 | 50.26 | 9.12 | 46.75 | 20.06 |
|  | NK1 | 17.43 | 21.17 | 48.97 | 50.10 | 99.17 | - |
|  | MLP | 91.22 | 92.46 | 99.22 | 66.79 | 99.53 | 98.86 |
| Normal Test set 2 | SNC | 31.58 | 28.61 | 32.82 | 8.02 | 47.69 | 16.25 |
|  | linear | 61.54 | 62.28 | 42.46 | 7.33 | 41.20 | 16.44 |
|  | NK1 | 17.26 | 18.61 | 74.60 | 45.49 | 99.21 | - |
|  | MLP | 90.43 | 91.39 | 98.92 | 65.73 | 99.78 | 98.79 |
| Normal Test set 3 | SNC | 31.49 | 28.75 | 45.15 | 6.61 | 48.19 | 25.62 |
|  | linear | 68.12 | 71.43 | 44.93 | 7.99 | 53.61 | 24.95 |
|  | NK1 | 23.67 | 23.21 | 47.74 | 46.00 | 99.46 | - |
|  | MLP | 94.07 | 94.23 | 98.88 | 67.51 | 99.87 | 98.78 |
| Normal Test set 4 | SNC | 38.37 | 43.16 | 31.20 | 7.49 | 51.50 | 16.36 |
|  | linear | 59.68 | 57.23 | 50.29 | 6.09 | 53.58 | 20.61 |
|  | NK1 | 20.02 | 19.29 | 80.06 | 52.03 | 99.52 | - |
|  | MLP | 89.92 | 89.44 | 99.02 | 61.74 | 99.07 | 98.38 |

Table 7: Full results of $\beta$-VAE on Dsprites for the main and normal test set settings.

|  |  | x | y | size | orient | shape | zs |
|---|---|---|---|---|---|---|---|
| normal test set 0 | SNC | 76.85 | 75.00 | 44.44 | 13.89 | 52.67 | 23.92 |
|  | linear | 47.59 | 49.17 | 54.77 | 11.84 | 84.57 | 48.70 |
|  | NK1 | 23.98 | 28.35 | 69.64 | 23.71 | 99.05 | - |
|  | MLP | 80.65 | 79.30 | 98.92 | 43.05 | 98.94 | 98.48 |
| normal test set 1 | SNC | 75.33 | 74.54 | 44.47 | 12.02 | 52.51 | 22.30 |
|  | linear | 59.16 | 60.37 | 46.89 | 14.22 | 73.57 | 33.34 |
|  | NK1 | 26.26 | 27.84 | 84.47 | 50.73 | 99.99 | - |
|  | MLP | 81.46 | 85.46 | 98.73 | 80.30 | 99.99 | 98.73 |
| normal test set 2 | SNC | 58.51 | 65.02 | 44.59 | 10.37 | 49.79 | 18.20 |
|  | linear | 73.12 | 55.32 | 44.70 | 10.66 | 65.07 | 31.50 |
|  | NK1 | 36.64 | 36.20 | 73.92 | 29.06 | 94.66 | - |
|  | MLP | 79.52 | 81.48 | 98.99 | 60.69 | 98.93 | 98.59 |
| normal test set 3 | SNC | 74.07 | 76.21 | 44.40 | 9.30 | 45.48 | 19.92 |
|  | linear | 59.95 | 51.65 | 45.22 | 8.39 | 68.82 | 31.01 |
|  | NK1 | 24.42 | 35.19 | 79.75 | 53.27 | 99.24 | - |
|  | MLP | 82.82 | 90.03 | 99.31 | 76.45 | 99.97 | 99.29 |
| normal test set 4 | SNC | 76.66 | 51.64 | 44.21 | 12.28 | 49.30 | 20.82 |
|  | linear | 59.30 | 75.21 | 43.98 | 9.03 | 66.13 | 28.84 |
|  | NK1 | 23.53 | 38.40 | 80.50 | 33.70 | 99.96 | - |
|  | MLP | 81.54 | 87.05 | 98.97 | 72.86 | 99.96 | 98.95 |

Table 8: Full results of $\beta$-TCVAE on Dsprites for the main and normal test set settings.

|  |  | x | y | size | orient | shape | NC |
|---|---|---|---|---|---|---|---|
| Normal Test set 0 | SNC | 11.28 | 17.09 | 45.07 | 4.49 | 43.69 | 18.85 |
|  | linear | 31.39 | 29.82 | 49.05 | 4.66 | 46.28 | 20.45 |
|  | NK1 | 41.93 | 22.33 | 33.99 | 19.36 | 93.18 | - |
|  | MLP | 76.29 | 77.66 | 97.36 | 33.98 | 98.58 | 96.29 |
| Normal Test set 1 | SNC | 10.81 | 7.93 | 46.25 | 3.96 | 46.74 | 22.18 |
|  | linear | 20.26 | 21.70 | 49.82 | 3.89 | 43.93 | 22.90 |
|  | NK1 | 22.60 | 40.56 | 35.44 | 18.87 | 93.65 | - |
|  | MLP | 74.98 | 74.87 | 96.95 | 32.41 | 97.45 | 95.15 |
| Normal Test set 2 | SNC | 11.81 | 17.05 | 47.09 | 4.13 | 43.17 | 20.49 |
|  | linear | 5.53 | 27.45 | 49.76 | 6.27 | 53.39 | 27.48 |
|  | NK1 | 44.08 | 42.64 | 39.35 | 23.61 | 94.90 | - |
|  | MLP | 78.16 | 78.72 | 98.43 | 42.24 | 98.96 | 97.72 |
| Normal Test set 3 | SNC | 16.38 | 17.62 | 48.49 | 5.78 | 47.36 | 22.66 |
|  | linear | 31.39 | 32.27 | 52.65 | 3.97 | 46.45 | 24.15 |
|  | NK1 | 25.68 | 20.32 | 34.40 | 18.53 | 95.26 | - |
|  | MLP | 76.09 | 77.75 | 97.86 | 36.38 | 98.55 | 96.82 |
| Normal Test set 4 | SNC | 7.60 | 9.13 | 48.08 | 3.38 | 44.32 | 21.36 |
|  | linear | 29.74 | 25.99 | 51.36 | 4.30 | 48.41 | 27.40 |
|  | NK1 | 44.79 | 46.11 | 37.80 | 23.25 | 92.10 | - |
|  | MLP | 74.00 | 75.37 | 97.00 | 37.73 | 98.55 | 96.03 |

Table 9: Full results of FactorVAE on Dsprites for the main and normal test set settings.

|  |  | x | y | size | orient | shape | NC |
|---|---|---|---|---|---|---|---|
| Normal Test set 0 | SNC | 47.10 | 12.54 | 27.42 | 9.11 | 43.62 | 13.83 |
|  | linear | 42.23 | 48.93 | 42.88 | 8.64 | 52.07 | 22.91 |
|  | NK1 | 5.90 | 56.56 | 48.59 | 7.91 | 75.46 | - |
|  | MLP | 72.14 | 68.87 | 75.58 | 32.03 | 95.87 | 73.99 |
| Normal Test set 1 | SNC | 13.76 | 8.47 | 29.44 | 7.54 | 40.12 | 11.44 |
|  | linear | 31.00 | 36.90 | 33.00 | 7.50 | 49.71 | 15.37 |
|  | NK1 | 10.05 | 37.85 | 67.24 | 9.31 | 80.02 | - |
|  | MLP | 72.75 | 74.11 | 57.29 | 29.83 | 97.91 | 56.84 |
| Normal Test set 2 | SNC | 10.13 | 6.04 | 33.26 | 8.11 | 53.70 | 16.78 |
|  | linear | 30.84 | 31.57 | 39.80 | 11.07 | 60.01 | 23.97 |
|  | NK1 | 9.97 | 5.95 | 33.05 | 8.07 | 54.04 | 16.67 |
|  | MLP | 70.26 | 74.17 | 78.93 | 55.02 | 98.96 | 78.51 |
| Normal Test set 3 | SNC | 13.31 | 48.36 | 43.50 | 4.87 | 38.62 | 17.35 |
|  | linear | 52.68 | 41.83 | 42.51 | 4.26 | 46.01 | 17.42 |
|  | NK1 | 56.76 | 5.44 | 43.97 | 7.50 | 85.22 | - |
|  | MLP | 66.55 | 75.06 | 74.94 | 20.91 | 94.07 | 72.75 |
| Normal Test set 4 | SNC | 42.76 | 15.96 | 31.59 | 8.10 | 46.75 | 14.95 |
|  | linear | 37.48 | 46.10 | 46.31 | 11.65 | 45.52 | 20.31 |
|  | NK1 | 6.37 | 29.60 | 59.92 | 9.25 | 80.02 | - |
|  | MLP | 70.27 | 73.46 | 88.22 | 50.11 | 98.99 | 87.70 |

Table 10: Full results of PartedVAE on Dsprites for the main and normal test set settings.

|  |  | x | y | size | orient | shape | NC |
|---|---|---|---|---|---|---|---|
| Normal Test set 0 | SNC | 19.43 | 39.12 | 46.53 | 5.94 | 39.56 | - |
|  | linear | 40.02 | 36.01 | 47.00 | 3.24 | 43.69 | 20.85 |
|  | NK1 | 23.91 | 10.08 | 42.01 | 4.63 | 72.54 | - |
|  | MLP | 71.62 | 71.87 | 68.06 | 3.58 | 64.98 | 50.67 |
| Normal Test set 1 | SNC | 19.43 | 39.12 | 46.53 | 5.94 | 39.56 | 18.11 |
|  | linear | 3.11 | 3.11 | 16.78 | 2.48 | 33.27 | 5.56 |
|  | NK1 | 23.91 | 10.08 | 42.01 | 4.63 | 72.54 | - |
|  | MLP | 3.14 | 3.11 | 16.52 | 2.46 | 33.39 | 5.43 |
| Normal Test set 2 | SNC | 19.43 | 39.12 | 46.53 | 5.94 | 39.56 | 18.11 |
|  | linear | 34.37 | 38.18 | 44.65 | 6.63 | 46.61 | 23.49 |
|  | NK1 | 9.70 | 47.66 | 47.17 | 7.46 | 72.25 | - |
|  | MLP | 63.27 | 70.20 | 61.73 | 13.32 | 88.56 | 58.16 |
| Normal Test set 3 | SNC | 7.94 | 4.48 | 29.43 | 9.32 | 48.59 | 14.17 |
|  | linear | 7.86 | 4.40 | 29.43 | 9.52 | 48.59 | 14.17 |
|  | NK1 | 9.70 | 47.66 | 47.17 | 7.46 | 72.25 | - |
|  | MLP | 16.77 | 67.65 | 57.61 | 15.13 | 81.59 | - |
| Normal Test set 4 | SNC | 43.74 | 21.89 | 37.34 | 7.95 | 48.38 | 15.66 |
|  | linear | 43.56 | 22.18 | 37.34 | 7.89 | 48.38 | 15.66 |
|  | NK1 | 5.48 | 30.74 | 51.88 | 5.21 | 80.36 | - |
|  | MLP | 71.56 | 72.21 | 76.74 | 26.32 | 90.54 | 72.10 |

Table 11: Full results of PartedVAE-semisupervised on Dsprites for the main and normal test set settings.

|  |  | x | y | size | orient | shape | NC |
|---|---|---|---|---|---|---|---|
| Normal Test set 0 | SNC | 7.51 | 8.11 | 29.11 | 3.04 | 37.47 | 11.79 |
|  | linear | 37.30 | 42.80 | 48.32 | 3.51 | 42.66 | 18.55 |
|  | NK1 | 71.77 | 67.55 | 77.71 | 39.14 | 98.89 | - |
|  | MLP | 81.40 | 82.05 | 92.76 | 46.80 | 99.07 | 92.13 |
| Normal Test set 1 | SNC | 8.14 | 8.15 | 22.10 | 3.07 | 37.62 | 8.61 |
|  | linear | 39.39 | 38.30 | 28.17 | 3.53 | 40.74 | 10.64 |
|  | NK1 | 66.54 | 66.77 | 79.80 | 39.02 | 98.79 | - |
|  | MLP | 80.93 | 80.02 | 87.71 | 44.05 | 98.93 | 87.01 |
| Normal Test set 2 | SNC | 8.35 | 7.53 | 24.05 | 3.25 | 37.67 | 9.12 |
|  | linear | 41.35 | 41.21 | 44.33 | 3.47 | 42.61 | 18.22 |
|  | NK1 | 64.68 | 71.90 | 83.36 | 40.52 | 98.92 | - |
|  | MLP | 82.55 | 82.89 | 94.37 | 47.91 | 98.73 | 93.49 |
| Normal Test set 3 | SNC | 8.61 | 7.80 | 26.70 | 3.22 | 38.92 | 10.73 |
|  | linear | 38.94 | 38.46 | 43.26 | 3.87 | 43.54 | 17.74 |
|  | NK1 | 67.53 | 71.59 | 80.87 | 40.75 | 98.92 | - |
|  | MLP | 82.14 | 83.36 | 92.53 | 48.07 | 98.97 | 91.87 |
| Normal Test set 4 | SNC | 7.28 | 6.28 | 21.78 | 3.18 | 38.30 | 8.30 |
|  | linear | 38.70 | 41.32 | 35.41 | 3.10 | 42.46 | 13.88 |
|  | NK1 | 71.17 | 73.76 | 81.51 | 39.35 | 98.89 | - |
|  | MLP | 81.78 | 82.90 | 89.26 | 47.07 | 98.87 | 88.51 |

Table 12: Full results of WeakDE on Dsprites for the main and normal test set settings.

| | | floor h | wall h | size | orient | shape | obj h | NC |
|---|---|---|---|---|---|---|---|---|
| Normal Test set 0 | SNC | 20.23 | 22.98 | 35.49 | 24.63 | 36.73 | 20.61 | 12.85 |
| | linear | 99.06 | 99.25 | 55.55 | 87.31 | 35.56 | 99.03 | 13.93 |
| | NK1 | 98.92 | 97.14 | 87.62 | 42.19 | 99.91 | 99.10 | - |
| | MLP | 100.00 | 100.00 | 100.00 | 99.92 | 100.00 | 100.00 | 100.00 |
| Normal Test set 1 | SNC | 21.86 | 16.99 | 16.37 | 15.77 | 41.45 | 17.30 | 7.11 |
| | linear | 99.00 | 99.12 | 15.72 | 96.39 | 64.16 | 99.05 | 8.39 |
| | NK1 | 96.53 | 98.18 | 94.16 | 70.73 | 98.91 | 99.00 | - |
| | MLP | 100.00 | 100.00 | 97.34 | 100.00 | 100.00 | 100.00 | 99.74 |
| Normal Test set 2 | SNC | 29.08 | 18.93 | 28.50 | 12.61 | 36.68 | 18.54 | 10.97 |
| | linear | 99.28 | 98.94 | 66.14 | 31.35 | 79.27 | 99.10 | 55.60 |
| | NK1 | 95.52 | 98.28 | 93.71 | 81.05 | 99.40 | 99.11 | - |
| | MLP | 100.00 | 100.00 | 99.63 | 99.68 | 100.00 | 100.00 | 99.63 |
| Normal Test set 3 | SNC | 25.06 | 18.19 | 36.35 | 19.88 | 35.68 | 21.74 | 14.70 |
| | linear | 99.19 | 99.34 | 44.92 | 88.59 | 53.89 | 99.00 | 26.22 |
| | NK1 | 98.58 | 98.92 | 84.52 | 75.17 | 99.99 | 99.39 | - |
| | MLP | 100.00 | 100.00 | 99.98 | 99.97 | 99.99 | 99.99 | 99.97 |
| Normal Test set 4 | SNC | 22.60 | 19.65 | 25.16 | 65.74 | 35.74 | 21.88 | 9.35 |
| | linear | 99.30 | 98.80 | 46.12 | 83.24 | 36.65 | 99.01 | 13.09 |
| | NK1 | 96.60 | 98.85 | 96.51 | 58.67 | 99.13 | 99.21 | - |
| | MLP | 100.00 | 100.00 | 100.00 | 98.87 | 100.00 | 100.00 | 100.00 |

Table 13: Full results of $\beta$-VAE on 3dshapes for the main and normal test set settings.

| | | floor h | wall h | size | orient | shape | obj h | NC |
|---|---|---|---|---|---|---|---|---|
| Normal Test set 0 | SNC | 31.69 | 20.30 | 36.03 | 20.79 | 39.68 | 21.05 | 14.58 |
| | linear | 99.13 | 99.08 | 59.18 | 82.33 | 29.42 | 99.03 | 17.38 |
| | NK1 | 85.91 | 95.27 | 92.73 | 65.54 | 99.37 | 97.91 | - |
| | MLP | 100.00 | 100.00 | 98.76 | 99.98 | 99.81 | 100.00 | 98.51 |
| Normal Test set 1 | SNC | 27.81 | 19.87 | 21.44 | 25.79 | 39.33 | 25.51 | 9.74 |
| | linear | 98.83 | 98.75 | 31.02 | 83.30 | 37.69 | 98.96 | 10.72 |
| | NK1 | 88.46 | 98.28 | 92.15 | 75.89 | 99.19 | 99.40 | - |
| | MLP | 100.00 | 100.00 | 99.72 | 99.98 | 100.00 | 100.00 | 99.72 |
| Normal Test set 2 | SNC | 21.71 | 18.87 | 28.32 | 43.90 | 34.91 | 20.41 | 12.09 |
| | linear | 98.87 | 99.10 | 31.58 | 73.53 | 35.15 | 99.00 | 9.22 |
| | NK1 | 97.43 | 98.97 | 94.61 | 72.15 | 98.87 | 98.85 | - |
| | MLP | 100.00 | 100.00 | 100.00 | 99.03 | 100.00 | 100.00 | 100.00 |
| Normal Test set 3 | SNC | 20.46 | 19.47 | 17.74 | 19.26 | 35.79 | 28.23 | 6.16 |
| | linear | 98.94 | - | 17.01 | 83.20 | 67.16 | 98.97 | 10.27 |
| | NK1 | 98.68 | 97.62 | 95.41 | 79.64 | 99.40 | 98.71 | - |
| | MLP | 100.00 | 100.00 | 99.71 | 99.99 | 100.00 | 100.00 | 99.71 |
| Normal Test set 4 | SNC | 24.25 | 18.61 | 22.31 | 20.17 | 42.96 | 20.95 | 10.65 |
| | linear | 98.96 | 99.06 | 18.62 | 83.30 | 70.90 | 99.03 | 12.27 |
| | NK1 | 93.09 | 95.87 | 99.13 | 47.21 | 99.84 | 97.66 | - |
| | MLP | 100.00 | 100.00 | 99.99 | 99.85 | 100.00 | 100.00 | 99.99 |

Table 14: Full results of $\beta$-TCVAE on 3dshapes for the main and normal test set settings.

|  |  | floor h | wall h | size | orient | shape | obj h | NC |
|---|---|---|---|---|---|---|---|---|
| Normal Test set 0 | SNC | 30.63 | 23.69 | 31.55 | 15.74 | 31.90 | 12.77 | 10.63 |
|  | linear | 98.94 | 98.97 | 38.43 | 38.27 | 31.99 | 74.98 | 11.52 |
|  | NK1 | 83.37 | 89.94 | 79.34 | 67.52 | 98.39 | 94.20 | - |
|  | MLP | 100.00 | 99.54 | 67.03 | 98.46 | 98.90 | 98.99 | 66.20 |
| Normal Test set 1 | SNC | 23.26 | 14.53 | 24.64 | 10.35 | 33.13 | 17.75 | 8.01 |
|  | linear | 96.79 | 99.02 | 18.76 | 57.48 | 54.74 | 86.12 | 9.14 |
|  | NK1 | 87.50 | 91.06 | 94.80 | 64.25 | 99.03 | 94.86 | - |
|  | MLP | 100.00 | 100.00 | 98.95 | 99.25 | 98.86 | 99.19 | 97.82 |
| Normal Test set 2 | SNC | 21.50 | 20.55 | 23.11 | 15.92 | 26.72 | 18.10 | 6.63 |
|  | linear | 99.03 | 99.04 | 37.24 | 49.19 | 36.76 | 90.54 | 10.63 |
|  | NK1 | 83.37 | 89.94 | 79.34 | 67.52 | 98.39 | 94.20 | - |
|  | MLP | 100.00 | 100.00 | 99.02 | 99.30 | 99.91 | 99.99 | 98.93 |
| Normal Test set 3 | SNC | 24.16 | 41.23 | 17.53 | 18.93 | 41.61 | 23.83 | 7.54 |
|  | linear | 99.08 | 99.09 | 17.62 | 52.32 | 50.72 | 76.24 | 8.70 |
|  | NK1 | 70.51 | 77.89 | 73.82 | 43.53 | 91.63 | 94.80 | - |
|  | MLP | 100.00 | 99.99 | 99.04 | 94.05 | 97.18 | 99.27 | 96.33 |
| Normal Test set 4 | SNC | 21.65 | 20.46 | 21.08 | 15.24 | 31.19 | 23.95 | 6.46 |
|  | linear | 95.49 | 93.47 | 18.97 | 31.85 | 38.35 | 75.64 | 5.68 |
|  | NK1 | 53.05 | 67.19 | 97.95 | 49.33 | 89.71 | 73.08 | - |
|  | MLP | 100.00 | 99.49 | 98.94 | 99.21 | 99.03 | 99.06 | 97.99 |

Table 15: Full results of FactorVAE on 3dshapes for the main and normal test set settings.

|  |  | floor h | wall h | size | orient | shape | obj h | NC |
|---|---|---|---|---|---|---|---|---|
| Normal Test set 0 | SNC | 99.86 | 37.53 | 38.02 | 99.53 | 93.45 | 82.14 | 35.10 |
|  | linear | 89.72 | 96.97 | 17.46 | 78.20 | 82.05 | 82.20 | 13.72 |
|  | NK1 | 26.44 | 94.22 | 18.01 | 8.04 | 13.02 | 61.88 | - |
|  | MLP | 99.07 | 98.86 | 90.97 | 99.07 | 99.03 | 98.94 | 90.13 |
| Normal Test set 1 | SNC | 7.50 | 6.68 | 30.74 | 8.29 | 44.40 | 13.48 | - |
|  | linear | 99.04 | 90.69 | 33.07 | 15.41 | 89.03 | 98.97 | 30.04 |
|  | NK1 | 14.24 | 90.43 | 20.57 | 9.57 | 17.77 | 56.55 | - |
|  | MLP | 98.86 | 99.37 | 98.94 | 95.95 | 99.12 | 99.70 | 98.07 |
| Normal Test set 2 | SNC | 9.97 | 5.95 | 33.05 | 8.07 | 54.04 | 16.67 | - |
|  | linear | 91.98 | 90.53 | 50.64 | 9.16 | 79.54 | 42.59 | 40.33 |
|  | NK1 | 10.40 | 45.82 | 80.42 | 6.82 | 15.06 | 56.19 | - |
|  | MLP | 98.87 | 99.04 | 96.26 | 30.39 | 98.95 | 98.84 | 95.29 |
| Normal Test set 3 | SNC | 30.85 | 31.22 | 28.99 | 99.52 | 66.82 | 34.19 | 19.56 |
|  | linear | 97.81 | 94.28 | 28.81 | 76.05 | 87.90 | 57.23 | 25.61 |
|  | NK1 | 10.40 | 45.82 | 80.42 | 6.82 | 15.06 | 56.19 | - |
|  | MLP | 98.98 | 98.34 | 98.12 | 99.29 | 99.11 | 99.05 | 97.32 |
| Normal Test set 4 | SNC | 99.88 | 47.20 | 55.89 | 99.34 | 97.51 | 67.04 | 54.05 |
|  | linear | 96.44 | 95.66 | 22.61 | 77.58 | 92.48 | 58.63 | 20.56 |
|  | NK1 | 26.44 | 94.22 | 18.01 | 8.04 | 13.02 | 61.88 | - |
|  | MLP | 98.93 | 99.10 | 98.98 | 99.05 | 99.01 | 99.07 | 97.99 |

Table 16: Full results of PartedVAE on 3dshapes for the main and normal test set settings.

|  |  | floor h | wall h | size | orient | shape | obj h | NC |
|---|---|---|---|---|---|---|---|---|
| Normal Test set 0 | SNC | 90.16 | 99.79 | 27.72 | 87.55 | 96.70 | 43.88 | 25.92 |
|  | linear | 61.00 | 62.00 | 41.00 | 8.00 | 99.00 | 51.00 | 40.00 |
|  | NK1 | 99.98 | 15.74 | 28.94 | 15.21 | 21.65 | 51.11 | - |
|  | MLP | 99.97 | 99.33 | 79.00 | 94.17 | 93.03 | 98.61 | 75.68 |
| Normal Test set 1 | SNC | 99.89 | 99.78 | 68.99 | 18.86 | 94.32 | 95.50 | 64.38 |
|  | linear | 81.55 | 86.23 | 82.90 | 44.33 | 98.64 | 78.13 | 82.02 |
|  | NK1 | 26.95 | 35.40 | 46.68 | 47.05 | 81.65 | 38.12 | - |
|  | MLP | 99.41 | 99.29 | 98.98 | 58.70 | 99.20 | 99.09 | 98.26 |
| Normal Test set 2 | SNC | 99.82 | 92.48 | 28.06 | 54.41 | 96.78 | 87.30 | 27.06 |
|  | linear | 91.99 | 82.74 | 38.46 | 50.64 | 98.75 | 82.02 | 37.57 |
|  | NK1 | 27.60 | 66.39 | 79.25 | 25.12 | 75.34 | 36.68 | - |
|  | MLP | 99.45 | 99.15 | 83.86 | 77.90 | 99.50 | 99.05 | 83.42 |
| Normal Test set 3 | SNC | 53.39 | 97.57 | 27.78 | 20.63 | 81.10 | 86.31 | 23.05 |
|  | linear | 98.94 | 85.54 | 27.05 | 8.22 | 85.31 | 82.89 | 22.84 |
|  | NK1 | 98.96 | 49.49 | 50.40 | 22.36 | 90.80 | 29.90 | - |
|  | MLP | 99.01 | 99.23 | 83.17 | 44.09 | 99.01 | 98.93 | 82.56 |
| Normal Test set 4 | SNC | 98.91 | 99.48 | 17.40 | 8.05 | 47.32 | 86.12 | 6.76 |
|  | linear | 69.56 | 81.85 | 32.41 | 13.94 | 62.67 | 75.35 | 19.28 |
|  | NK1 | 28.90 | 27.48 | 51.00 | 16.60 | 93.09 | 35.97 | - |
|  | MLP | 99.05 | 98.99 | 54.16 | 19.93 | 96.11 | 98.41 | 52.52 |

Table 17: Full results of PartedVAE-semisupervised on 3dshapes for the main and normal test set settings.

|  |  | floor h | wall h | size | orient | shape | obj h | NC |
|---|---|---|---|---|---|---|---|---|
| Normal Test set 0 | SNC | 19.50 | 22.68 | 17.03 | 9.62 | 27.94 | 19.02 | 4.98 |
|  | linear | 99.22 | 99.31 | 13.61 | 15.92 | 35.06 | 99.02 | 4.77 |
|  | NK1 | 98.97 | 99.05 | 88.36 | 98.51 | 99.02 | 98.87 | - |
|  | MLP | 100.00 | 100.00 | 99.92 | 99.18 | 100.00 | 100.00 | 99.92 |
| Normal Test set 1 | SNC | 22.14 | 23.58 | 17.98 | 9.43 | 50.12 | 19.52 | 9.04 |
|  | linear | 98.80 | 98.96 | 32.86 | 23.15 | 70.65 | 96.92 | 23.18 |
|  | NK1 | 98.25 | 98.94 | 95.03 | 91.48 | 99.09 | 98.47 | - |
|  | MLP | 100.00 | 100.00 | 98.90 | 98.61 | 99.97 | 99.92 | 98.87 |
| Normal Test set 2 | SNC | 15.70 | 23.55 | 18.35 | 8.41 | 31.98 | 28.50 | 6.16 |
|  | linear | 99.10 | 99.24 | 15.56 | 19.52 | 36.15 | 98.96 | 6.76 |
|  | NK1 | 98.98 | 99.17 | 88.31 | 97.94 | 98.94 | 98.98 | - |
|  | MLP | 100.00 | 100.00 | 99.46 | 99.74 | 100.00 | 100.00 | 99.46 |
| Normal Test set 3 | SNC | 17.46 | 20.01 | 18.14 | 13.83 | 38.22 | 20.15 | 6.92 |
|  | linear | 99.18 | 99.31 | 28.04 | 90.54 | 45.35 | 99.02 | 10.36 |
|  | NK1 | 98.89 | 99.05 | 98.88 | 88.39 | 99.36 | 98.62 | - |
|  | MLP | 100.00 | 100.00 | 99.74 | 99.24 | 100.00 | 100.00 | 99.74 |
| Normal Test set 4 | SNC | 18.07 | 16.72 | 21.09 | 13.43 | 54.16 | 17.62 | 11.16 |
|  | linear | 99.59 | 99.26 | 97.40 | 90.49 | 99.48 | 99.29 | 96.89 |
|  | NK1 | 98.97 | 99.32 | 98.80 | 95.86 | 99.21 | 99.05 | - |
|  | MLP | 100.00 | 100.00 | 99.99 | 99.78 | 100.00 | 100.00 | 99.99 |

Table 18: Full results of WeakDE on 3dshapes for the main and normal test set settings.

|  |  | hor | vert | size | obj h | shape | cam he | bg h | NC |
|---|---|---|---|---|---|---|---|---|---|
| Normal Test set 0 | SNC | 5.30 | 5.24 | 66.06 | 73.80 | 26.15 | 94.84 | 54.61 | 17.39 |
|  | linear | 17.63 | 11.68 | 77.79 | 76.17 | 36.94 | 99.71 | 48.30 | 28.21 |
|  | NK1 | 66.18 | 40.06 | 92.69 | 70.65 | 74.81 | 88.90 | 50.87 | - |
|  | MLP | 86.51 | 70.23 | 98.92 | 99.82 | 88.28 | 100.00 | 99.99 | 87.61 |
| Normal Test set 1 | SNC | 6.26 | 4.45 | 66.69 | 78.82 | 27.08 | 74.31 | 52.44 | 18.36 |
|  | linear | 16.87 | 9.24 | 76.29 | 88.91 | 34.92 | 100.00 | 45.30 | 25.93 |
|  | NK1 | 59.51 | 40.68 | 91.87 | 69.46 | 71.61 | 91.04 | 70.78 | - |
|  | MLP | 86.01 | 66.49 | 98.96 | 99.90 | 87.26 | 100.00 | 99.83 | 86.56 |
| Normal Test set 2 | SNC | 5.92 | 5.63 | 59.77 | 57.65 | 23.22 | 87.11 | 36.20 | 14.21 |
|  | linear | 15.54 | 14.62 | 71.78 | 73.45 | 34.85 | 99.89 | 36.64 | 24.08 |
|  | NK1 | 60.73 | 37.18 | 94.33 | 84.45 | 68.74 | 85.02 | 79.00 | - |
|  | MLP | 85.17 | 68.24 | 98.68 | 99.58 | 84.47 | 100.00 | 99.99 | 83.71 |
| Normal Test set 3 | SNC | 5.77 | 4.95 | 62.93 | 72.61 | 27.47 | 73.22 | 55.50 | 17.16 |
|  | linear | 15.54 | 14.62 | 71.78 | 73.45 | 34.85 | 99.89 | 36.64 | 24.08 |
|  | NK1 | 59.92 | 40.15 | 93.69 | 76.82 | 69.83 | 95.18 | 58.62 | - |
|  | MLP | 87.32 | 67.92 | 98.90 | 99.74 | 84.63 | 100.00 | 100.00 | 83.94 |
| Normal Test set 4 | SNC | 6.25 | 4.59 | 56.47 | 62.55 | 27.04 | 58.91 | 40.52 | 15.17 |
|  | linear | 14.15 | 8.50 | 64.75 | 97.15 | 33.86 | 99.08 | 44.15 | 21.45 |
|  | NK1 | 62.05 | 40.96 | 94.50 | 81.50 | 68.85 | 97.42 | 73.09 | - |
|  | MLP | 86.68 | 68.55 | 98.88 | 99.86 | 84.24 | 100.00 | 99.99 | 83.54 |

Table 19: Full results of $\beta$-VAE on MPI3D for the main and normal test set settings.

|  |  | hor | vert | size | obj h | shape | cam he | bg h | NC |
|---|---|---|---|---|---|---|---|---|---|
| Normal Test set 0 | SNC | 5.13 | 4.14 | 65.64 | 70.81 | 23.31 | 67.86 | 55.50 | 14.34 |
|  | linear | 13.54 | 10.34 | 75.53 | 70.25 | 32.45 | 99.84 | 40.92 | 23.82 |
|  | NK1 | 56.75 | 40.24 | 92.83 | 80.80 | 66.90 | 91.88 | 46.96 | - |
|  | MLP | 85.59 | 66.92 | 98.79 | 99.44 | 86.87 | 100.00 | 100.00 | 86.06 |
| Normal Test set 1 | SNC | 5.91 | 5.60 | 69.62 | 71.87 | 27.90 | 67.77 | 55.60 | 18.94 |
|  | linear | 17.20 | 12.70 | 76.87 | 75.50 | 35.48 | 100.00 | 46.89 | 26.01 |
|  | NK1 | 56.99 | 40.43 | 92.79 | 80.58 | 66.93 | 91.78 | 46.76 | - |
|  | MLP | 86.95 | 67.02 | 98.92 | 99.73 | 86.77 | 100.00 | 100.00 | 86.10 |
| Normal Test set 2 | SNC | 6.80 | 4.53 | 57.17 | 39.40 | 24.66 | 77.33 | 99.99 | 13.77 |
|  | linear | 15.48 | 9.25 | 65.72 | 43.77 | 31.83 | 99.54 | 100.00 | 20.39 |
|  | NK1 | 58.99 | 40.16 | 95.22 | 74.86 | 65.18 | 86.65 | 35.03 | - |
|  | MLP | 86.18 | 65.38 | 97.93 | 99.38 | 81.62 | 100.00 | 0.00 | 80.38 |
| Normal Test set 3 | SNC | 6.25 | 5.40 | 59.49 | 75.47 | 26.99 | 76.81 | 99.99 | 16.53 |
|  | linear | 13.13 | 7.52 | 66.07 | 74.07 | 34.60 | 99.90 | 99.99 | 22.44 |
|  | NK1 | 64.93 | 40.40 | 93.30 | 74.53 | 66.91 | 84.87 | 35.26 | - |
|  | MLP | 87.72 | 69.01 | 98.36 | 99.92 | 82.72 | 100.00 | 0.00 | 81.69 |
| Normal Test set 4 | SNC | 5.82 | 4.47 | 64.64 | 77.51 | 23.02 | 68.20 | 55.64 | 14.48 |
|  | linear | 15.39 | 9.03 | 70.90 | 76.98 | 34.63 | 99.55 | 48.02 | 23.89 |
|  | NK1 | 45.27 | 29.36 | 88.51 | 74.28 | 56.42 | 89.65 | 45.42 | - |
|  | MLP | 67.18 | 48.54 | 93.79 | 98.46 | 68.25 | 100.00 | 99.90 | 65.03 |

Table 20: Full results of $\beta$-TCVAE on MPI3D for the main and normal test set settings.

|  |  | hor | vert | size | obj h | shape | cam he | bg h | NC |
|---|---|---|---|---|---|---|---|---|---|
| Normal Test set 0 | SNC | 6.14 | 4.98 | 64.12 | 40.64 | 24.57 | 68.75 | 46.87 | 15.79 |
|  | linear | 12.66 | 8.85 | 63.69 | 52.95 | 36.11 | 99.10 | 36.79 | 23.72 |
|  | NK1 | 47.85 | 29.58 | 89.10 | 74.60 | 63.35 | 81.47 | 59.91 | - |
|  | MLP | 77.95 | 59.73 | 97.43 | 98.88 | 80.46 | 100.00 | 99.83 | 78.84 |
| Normal Test set 1 | SNC | 5.15 | 4.48 | 56.76 | 69.91 | 26.81 | 75.57 | 52.86 | 15.10 |
|  | linear | 6.30 | 9.78 | 61.18 | 76.65 | 35.10 | 99.33 | 35.35 | 21.39 |
|  | NK1 | 54.76 | 32.60 | 91.11 | 66.24 | 65.88 | 80.98 | 62.50 | - |
|  | MLP | 80.17 | 62.57 | 98.12 | 99.51 | 83.14 | 100.00 | 99.88 | 82.00 |
| Normal Test set 2 | SNC | 5.23 | 4.63 | 64.89 | 34.89 | 26.00 | 70.51 | 50.20 | 16.53 |
|  | linear | 15.99 | 10.08 | 75.54 | 46.83 | 41.87 | 99.68 | 45.27 | 31.97 |
|  | NK1 | 51.14 | 33.54 | 89.47 | 75.24 | 69.38 | 81.07 | 58.45 | - |
|  | MLP | 79.81 | 60.92 | 98.25 | 98.73 | 84.47 | 100.00 | 99.79 | 83.31 |
| Normal Test set 3 | SNC | 6.17 | 4.47 | 61.02 | 35.92 | 24.58 | 74.69 | 38.31 | 15.23 |
|  | linear | 11.41 | 9.66 | 69.75 | 77.15 | 39.51 | 98.97 | 38.68 | 27.03 |
|  | NK1 | 51.61 | 34.31 | 90.53 | 77.10 | 66.95 | 85.15 | 78.39 | - |
|  | MLP | 81.20 | 62.05 | 97.99 | 99.09 | 82.61 | 100.00 | 99.13 | 81.28 |
| Normal Test set 4 | SNC | 6.06 | 4.18 | 61.39 | 47.24 | 27.46 | 67.78 | 55.04 | 17.39 |
|  | linear | 13.21 | 8.51 | 65.71 | 57.53 | 36.77 | 99.11 | 46.84 | 24.16 |
|  | NK1 | 52.83 | 32.99 | 90.80 | 70.03 | 61.81 | 84.48 | 53.82 | - |
|  | MLP | 80.62 | 61.44 | 97.49 | 98.91 | 80.69 | 100.00 | 99.98 | 79.16 |

Table 21: Full results of FactorVAE on MPI3D for the main and normal test set settings.

|  |  | hor | vert | size | obj h | shape | cam he | bg h | NC |
|---|---|---|---|---|---|---|---|---|---|
| Normal Test set 0 | SNC | 10.13 | 6.90 | 59.61 | 20.70 | 85.20 | 79.62 | 21.03 | 50.95 |
|  | linear | 13.61 | 5.03 | 74.11 | 30.09 | 100.00 | 99.84 | 28.68 | 19.89 |
|  | NK1 | 18.90 | 8.51 | 74.45 | 76.52 | 100.00 | 92.26 | 32.14 | - |
|  | MLP | 38.36 | 26.28 | 81.37 | 93.46 | 100.00 | 99.97 | 37.50 | 31.20 |
| Normal Test set 1 | SNC | 9.03 | 6.75 | 59.86 | 23.50 | 99.46 | 89.06 | 21.09 | 59.42 |
|  | linear | 13.81 | 4.39 | 72.60 | 30.92 | 100.00 | 98.54 | 28.38 | 19.38 |
|  | NK1 | 16.40 | 6.67 | 76.40 | 71.57 | 100.00 | 94.37 | 31.05 | - |
|  | MLP | 35.75 | 23.08 | 81.95 | 88.11 | 100.00 | 99.87 | 35.97 | 30.19 |
| Normal Test set 2 | SNC | 7.51 | 4.23 | 69.29 | 20.73 | 95.12 | 89.54 | 21.89 | 65.55 |
|  | linear | 12.42 | 4.52 | 75.20 | 32.37 | 100.00 | 98.98 | 28.73 | 19.95 |
|  | NK1 | 15.15 | 9.09 | 73.99 | 72.14 | 100.00 | 83.84 | 29.93 | - |
|  | MLP | 32.55 | 21.78 | 79.78 | 84.59 | 100.00 | 99.80 | 34.82 | 28.86 |
| Normal Test set 3 | SNC | 8.16 | 5.18 | 66.09 | 23.87 | 99.91 | 99.97 | 21.12 | 66.01 |
|  | linear | 11.58 | 5.26 | 73.16 | 34.27 | 100.00 | 99.40 | 27.87 | 19.20 |
|  | NK1 | 15.15 | 9.09 | 73.99 | 72.14 | 100.00 | 83.84 | 29.93 | - |
|  | MLP | 29.91 | 21.00 | 80.05 | 86.00 | 100.00 | 99.99 | 33.08 | 26.50 |
| Normal Test set 4 | SNC | 7.27 | 4.43 | 72.89 | 21.18 | 99.93 | 66.75 | 24.29 | 72.84 |
|  | linear | 12.72 | 5.04 | 74.30 | 32.78 | 100.00 | 70.82 | 29.77 | 21.37 |
|  | NK1 | 14.64 | 7.27 | 77.84 | 84.96 | 100.00 | 74.73 | 31.71 | - |
|  | MLP | 32.78 | 24.02 | 80.31 | 91.45 | 100.00 | 84.24 | 34.44 | 28.60 |

Table 22: Full results of PartedVAE on MPI3D for the main and normal test set settings.

|  |  | hor | vert | size | obj h | shape | cam he | bg h | NC |
|---|---|---|---|---|---|---|---|---|---|
| Normal Test set 0 | SNC | 6.72 | 3.87 | 59.61 | 22.89 | 19.24 | 86.08 | 34.16 | 11.75 |
|  | linear | 11.16 | 5.91 | 63.43 | 33.39 | 24.05 | 99.19 | 35.72 | 14.95 |
|  | NK1 | 23.80 | 16.78 | 74.27 | 71.11 | 31.55 | 100.00 | 50.64 | - |
|  | MLP | 25.23 | 16.15 | 74.58 | 72.87 | 31.49 | 100.00 | 51.22 | 23.16 |
| Normal Test set 1 | SNC | 5.11 | 4.01 | 64.20 | 26.78 | 18.82 | 77.92 | 53.51 | 11.86 |
|  | linear | 13.61 | 5.10 | 71.16 | 38.57 | 26.76 | 99.79 | 56.38 | 17.84 |
|  | NK1 | 25.79 | 12.96 | 79.32 | 84.73 | 33.45 | 100.00 | 60.71 | - |
|  | MLP | 29.60 | 18.78 | 79.98 | 86.22 | 33.07 | 100.00 | 62.59 | 26.30 |
| Normal Test set 2 | SNC | 5.54 | 5.45 | 67.48 | 31.15 | 19.64 | 58.11 | 56.64 | 13.64 |
|  | linear | 11.16 | 7.16 | 72.97 | 35.26 | 26.55 | 99.59 | 50.12 | 18.07 |
|  | NK1 | 27.81 | 19.01 | 80.15 | 68.04 | 34.17 | 99.89 | 89.02 | - |
|  | MLP | 31.35 | 22.00 | 81.21 | 91.17 | 35.05 | 99.88 | 99.15 | 28.98 |
| Normal Test set 3 | SNC | 7.27 | 4.43 | 69.31 | 22.95 | 20.91 | 75.50 | 47.48 | 14.97 |
|  | linear | 13.15 | 6.35 | 74.10 | 32.70 | 29.05 | 100.00 | 84.06 | 20.00 |
|  | NK1 | 26.60 | 16.82 | 80.10 | 70.85 | 36.12 | 100.00 | 98.94 | - |
|  | MLP | 34.88 | 21.69 | 81.56 | 97.47 | 36.37 | 100.00 | 98.93 | 30.72 |
| Normal Test set 4 | SNC | 7.50 | 4.95 | 57.62 | 28.21 | 19.86 | 79.57 | 35.90 | 11.41 |
|  | linear | 11.22 | 5.62 | 67.17 | 30.98 | 26.43 | 99.64 | 38.44 | 16.83 |
|  | NK1 | 27.89 | 15.32 | 80.01 | 80.55 | 33.47 | 100.00 | 54.95 | - |
|  | MLP | 31.72 | 20.49 | 80.47 | 87.28 | 33.94 | 100.00 | 54.83 | 27.71 |

Table 23: Full results of PartedVAE-semisupervised on MPI3D for the main and normal test set settings.

|  |  | hor | vert | size | obj h | shape | cam he | bg h | NC |
|---|---|---|---|---|---|---|---|---|---|
| Normal Test set 0 | SNC | 0.57 | 3.84 | 52.53 | 20.78 | 18.40 | 71.63 | 35.50 | 9.66 |
|  | linear | 15.81 | 17.02 | 58.85 | 33.60 | 25.14 | 100.00 | 62.21 | 15.30 |
|  | NK1 | 70.17 | 48.03 | 88.73 | 82.23 | 53.53 | 100.00 | 93.98 | - |
|  | MLP | 76.40 | 60.47 | 90.95 | 98.67 | 60.03 | 100.00 | 99.99 | 56.00 |
| Normal Test set 1 | SNC | 5.16 | 4.72 | 52.89 | 19.39 | 19.53 | 72.53 | 35.00 | 10.75 |
|  | linear | 18.03 | 16.50 | 65.62 | 25.35 | 27.76 | 100.00 | 79.91 | 17.64 |
|  | NK1 | 75.25 | 53.43 | 89.21 | 84.39 | 55.02 | 100.00 | 98.85 | - |
|  | MLP | 77.80 | 60.68 | 92.24 | 97.39 | 58.37 | 100.00 | 100.00 | 55.17 |
| Normal Test set 2 | SNC | 6.59 | 4.96 | 54.00 | 21.59 | 18.64 | 69.29 | 35.13 | 10.30 |
|  | linear | 15.14 | 18.09 | 57.60 | 46.99 | 24.25 | 100.00 | 68.84 | 14.36 |
|  | NK1 | 71.32 | 49.02 | 85.29 | 90.70 | 52.83 | 99.92 | 99.00 | - |
|  | MLP | 75.79 | 59.77 | 91.18 | 98.35 | 58.47 | 100.00 | 99.98 | 54.64 |
| Normal Test set 3 | SNC | 5.67 | 4.88 | 52.95 | 19.65 | 19.62 | 69.05 | 35.11 | 10.64 |
|  | linear | 15.55 | 15.67 | 58.43 | 37.06 | 24.38 | 100.00 | 66.96 | 14.43 |
|  | NK1 | 65.89 | 41.08 | 88.59 | 86.18 | 55.12 | 100.00 | 95.88 | - |
|  | MLP | 73.11 | 56.72 | 91.00 | 98.74 | 59.10 | 100.00 | 100.00 | 55.35 |
| Normal Test set 4 | SNC | 7.34 | 4.77 | 50.52 | 20.14 | 18.68 | 68.40 | 37.05 | 9.30 |
|  | linear | 16.50 | 15.55 | 59.31 | 36.19 | 26.50 | 100.00 | 69.90 | 15.76 |
|  | NK1 | 63.21 | 45.49 | 88.87 | 83.88 | 55.06 | 100.00 | 98.30 | - |
|  | MLP | 74.64 | 58.74 | 90.68 | 98.68 | 59.65 | 100.00 | 100.00 | 55.68 |

Table 24: Full results of WeakDE on MPI3D for the main and normal test set settings.

|  |  | x | y | size | orient | shape | NC |
|---|---|---|---|---|---|---|---|
| NC 0-0 | CG linear | 30.14 | 19.38 | 7.91 | 2.64 | 0.01 | 0.00 |
|  | CG | 95.77 | 93.95 | 7.03 | 10.84 | 78.52 | 5.08 |
| NC 5-2 | CG linear | 21.06 | 21.44 | 18.52 | 8.38 | 8.94 | 0.00 |
|  | CG | 44.42 | 51.18 | 0.01 | 76.80 | 96.88 | 0.00 |
| NC 2-1 | CG linear | 3.12 | 3.12 | 0.00 | 2.50 | 0.00 | 0.00 |
|  | CG | 89.07 | 87.75 | 5.55 | 38.73 | 99.69 | 5.49 |
| NC 3-1 | CG linear | 86.13 | 30.04 | 0.00 | 9.85 | 2.16 | 0.00 |
|  | CG | 98.44 | 97.87 | 8.42 | 73.54 | 99.94 | 8.37 |
| NC 3-2 | CG linear | 22.47 | 20.41 | 0.00 | 9.40 | 9.70 | 0.00 |
|  | CG | 79.55 | 84.32 | 0.00 | 92.19 | 99.39 | 0.00 |

Table 25: Full results of $\beta$-VAE on Dsprites for the compositional generalization setting.

|  |  | x | y | size | orient | shape | NC |
|---|---|---|---|---|---|---|---|
| NC 0-0 | CG linear | 31.96 | 38.55 | 0.55 | 2.96 | 0.00 | 0.00 |
|  | CG | 91.24 | 96.66 | 0.00 | 6.40 | 19.02 | 0.00 |
| NC 5-2 | CG linear | 23.32 | 22.40 | 0.00 | 21.37 | 0.00 | 0.00 |
|  | CG | 22.06 | 19.77 | 0.00 | 40.17 | 95.48 | 0.00 |
| NC 2-1 | CG linear | 65.64 | 66.06 | 0.42 | 15.96 | 26.21 | 0.01 |
|  | CG | 97.68 | 95.48 | 0.00 | 78.81 | 99.99 | 0.00 |
| NC 3-1 | CG linear | 99.54 | 99.39 | 0.00 | 82.19 | 100.00 | 0.00 |
|  | CG | 99.54 | 99.39 | 0.00 | 82.19 | 100.00 | 0.00 |
| NC 3-2 | CG linear | 22.60 | 28.62 | 0.06 | 5.32 | 63.63 | 0.02 |
|  | CG | 38.94 | 51.74 | 0.00 | 94.16 | 100.00 | 0.00 |

Table 26: Full results of $\beta$-TCVAE on Dsprites for the compositional generalization setting.

|  |  | x | y | size | orient | shape | NC |
|---|---|---|---|---|---|---|---|
| NC 0-0 | CG linear | 37.29 | 29.55 | 2.13 | 2.91 | 0.00 | 0.00 |
|  | CG | 96.60 | 95.51 | 0.37 | 7.52 | 31.49 | 0.00 |
| NC 5-2 | CG linear | 33.84 | 35.84 | 0.00 | 7.23 | 6.25 | 0.00 |
|  | CG | 34.67 | 20.87 | 0.20 | 46.77 | 67.82 | 0.20 |
| NC 2-1 | CG linear | 26.98 | 27.80 | 11.16 | 7.22 | 57.42 | 1.57 |
|  | CG | 26.98 | 27.80 | 11.16 | 7.22 | 57.42 | 1.57 |
| NC 3-1 | CG linear | 24.69 | 17.07 | 2.32 | 4.78 | 31.77 | 0.00 |
|  | CG | 89.27 | 85.19 | 30.59 | 28.76 | 98.82 | 30.11 |
| NC 3-2 | CG linear | 23.10 | 17.11 | 0.00 | 6.23 | 46.23 | 0.00 |
|  | CG | 41.77 | 39.57 | 0.00 | 64.69 | 97.54 | 0.00 |

Table 27: Full results of FactorVAE on Dsprites for the compositional generalization setting.

|  |  | x | y | size | orient | shape | NC |
|---|---|---|---|---|---|---|---|
| NC 0-0 | CG linear | 8.78 | 25.76 | 15.76 | 2.47 | 2.52 | 0.00 |
|  | CG | 99.44 | 99.14 | 0.01 | 2.48 | 0.00 | 0.00 |
| NC 5-2 | CG linear | 19.78 | 21.52 | 0.24 | 16.12 | 1.33 | 0.00 |
|  | CG | 16.81 | 20.48 | 0.18 | 36.63 | 60.67 | 0.00 |
| NC 2-1 | CG linear | 50.62 | 49.59 | 0.02 | 26.67 | 98.71 | 0.00 |
|  | CG | 93.12 | 90.44 | 0.00 | 74.15 | 95.71 | 0.00 |
| NC 3-1 | CG linear | 31.28 | 39.57 | 2.21 | 5.88 | 23.20 | 0.00 |
|  | CG | 86.59 | 86.15 | 2.90 | 33.01 | 98.21 | 2.23 |
| NC 3-2 | CG linear | 25.13 | 25.57 | 0.00 | 10.81 | 2.38 | 0.00 |
|  | CG | 27.62 | 22.72 | 0.00 | 13.23 | 60.12 | 0.00 |

Table 28: Full results of PartedVAE on Dsprites for the compositional generalization setting.

|        |           | x     | y     | size  | orient | shape | NC   |
|--------|-----------|-------|-------|-------|--------|-------|------|
| NC 0-0 | CG linear | 43.96 | 26.48 | 13.41 | 2.95   | 7.58  | 0.00 |
|        | CG        | 80.95 | 87.81 | 1.19  | 4.81   | 65.34 | 0.07 |
| NC 5-2 | CG linear | 19.61 | 19.25 | 0.00  | 8.70   | 0.00  | 0.00 |
|        | CG        | 15.49 | 20.46 | 0.07  | 8.85   | 14.52 | 0.00 |
| NC 2-1 | CG linear | 53.81 | 69.43 | 1.07  | 15.81  | 39.05 | 0.00 |
|        | CG        | 94.84 | 93.81 | 0.23  | 50.94  | 99.83 | 0.23 |
| NC 3-1 | CG linear | 42.65 | 52.21 | 3.89  | 11.40  | 53.01 | 0.00 |
|        | CG        | 87.30 | 83.14 | 4.55  | 29.54  | 92.29 | 2.94 |
| NC 3-2 | CG linear | 25.22 | 25.84 | 0.19  | 7.90   | 15.72 | 0.00 |
|        | CG        | 21.01 | 25.08 | 0.02  | 14.37  | 53.86 | 0.00 |

Table 29: Full results of PartedVAE-semisupervised on Dsprites for the compositional generalization setting.

|        |           | x     | y     | size  | orient | shape  | NC   |
|--------|-----------|-------|-------|-------|--------|--------|------|
| NC 0-0 | CG linear | 67.06 | 55.46 | 28.83 | 2.75   | 0.00   | 0.00 |
|        | CG        | 95.00 | 93.37 | 4.57  | 13.08  | 13.14  | 0.54 |
| NC 5-2 | CG linear | 30.97 | 30.00 | 0.08  | 2.95   | 0.00   | 0.00 |
|        | CG        | 28.75 | 32.85 | 0.00  | 42.50  | 65.22  | 0.00 |
| NC 2-1 | CG linear | 45.72 | 45.86 | 15.60 | 3.80   | 0.24   | 0.00 |
|        | CG        | 90.76 | 91.81 | 3.38  | 50.05  | 100.00 | 3.38 |
| NC 3-1 | CG linear | 32.40 | 39.59 | 5.65  | 3.82   | 0.43   | 0.01 |
|        | CG        | 89.90 | 90.93 | 2.35  | 50.28  | 99.92  | 2.33 |
| NC 3-2 | CG linear | 32.24 | 35.32 | 2.93  | 3.03   | 16.48  | 0.00 |
|        | CG        | 50.35 | 49.95 | 0.00  | 60.74  | 97.33  | 0.00 |

Table 30: Full results of WeakDE on Dsprites for the compositional generalization setting.

|        |           | floor h | wall h | size  | orient | obj h  | shape  | NC    |
|--------|-----------|---------|--------|-------|--------|--------|--------|-------|
| NC 0-0 | CG linear | 99.25   | 99.03  | 48.31 | 86.55  | 0.00   | 96.83  | 0.00  |
|        | CG        | 100.00  | 100.00 | 46.61 | 100.00 | 52.34  | 99.89  | 4.97  |
| NC 5-2 | CG linear | 100.00  | 100.00 | 0.00  | 92.12  | 100.00 | 100.00 | 0.00  |
|        | CG        | 100.00  | 100.00 | 0.00  | 100.00 | 100.00 | 100.00 | 0.00  |
| NC 2-1 | CG linear | 100.00  | 99.87  | 0.00  | 64.17  | 16.77  | 100.00 | 0.00  |
|        | CG        | 100.00  | 100.00 | 38.27 | 99.66  | 79.52  | 100.00 | 32.94 |
| NC 3-1 | CG linear | 99.41   | 99.70  | 0.00  | 90.09  | 41.61  | 100.00 | 0.00  |
|        | CG        | 100.00  | 100.00 | 47.07 | 100.00 | 15.80  | 100.00 | 0.00  |
| NC 3-2 | CG linear | 99.95   | 99.93  | 15.86 | 90.67  | 0.00   | 99.33  | 0.00  |
|        | CG        | 100.00  | 100.00 | 0.00  | 99.77  | 100.00 | 100.00 | 0.00  |
| NC 7-3 | CG linear | 97.30   | 96.90  | 59.35 | 78.43  | 0.00   | 100.00 | 0.00  |
|        | CG        | 100.00  | 99.88  | 43.79 | 83.65  | 93.99  | 100.00 | 37.78 |
| NC 0-3 | CG linear | 99.59   | 99.81  | 0.00  | 89.38  | 63.08  | 68.14  | 0.00  |
|        | CG        | 100.00  | 100.00 | 53.88 | 100.00 | 44.76  | 67.69  | 14.41 |
| NC 7-1 | CG linear | 97.33   | 98.96  | 11.47 | 94.43  | 0.00   | 100.00 | 0.00  |
|        | CG        | 100.00  | 100.00 | 59.56 | 100.00 | 79.87  | 100.00 | 49.19 |

Table 31: Full results of $\beta$-VAE on 3dshapes for the compositional generalization setting.

|  |  | floor h | wall h | size | orient | obj h | shape | NC |
|---|---|---|---|---|---|---|---|---|
| NC 0-0 | CG linear | 93.59 | 95.03 | 45.30 | 90.57 | 76.21 | 76.68 | 26.85 |
|  | CG | 99.11 | 100.00 | 45.35 | 99.96 | 49.21 | 80.89 | 0.03 |
| NC 5-2 | CG linear | 99.12 | 98.65 | 1.41 | 97.91 | 95.38 | 90.47 | 1.41 |
|  | CG | 100.00 | 100.00 | 0.00 | 100.00 | 100.00 | 93.93 | 0.00 |
| NC 2-1 | CG linear | 100.00 | 100.00 | 0.00 | 88.60 | 74.31 | 99.99 | 0.00 |
|  | CG | 100.00 | 100.00 | 43.31 | 99.95 | 73.91 | 100.00 | 36.43 |
| NC 3-1 | CG linear | 99.85 | 99.88 | 0.00 | 83.13 | 84.91 | 100.00 | 0.00 |
|  | CG | 100.00 | 100.00 | 12.79 | 100.00 | 84.64 | 100.00 | 0.00 |
| NC 3-2 | CG linear | 99.04 | 99.73 | 0.00 | 85.46 | 99.99 | 99.87 | 0.00 |
|  | CG | 100.00 | 100.00 | 0.00 | 99.69 | 100.00 | 99.93 | 0.00 |
| NC 7-3 | CG linear | 99.99 | 99.21 | 8.26 | 83.51 | 25.83 | 100.00 | 2.92 |
|  | CG | 100.00 | 99.99 | 39.67 | 97.91 | 90.04 | 100.00 | 30.63 |
| NC 0-3 | CG linear | 99.94 | 100.00 | 0.04 | 87.69 | 4.49 | 69.07 | 0.00 |
|  | CG | 100.00 | 100.00 | 22.02 | 100.00 | 53.25 | 81.16 | 2.39 |
| NC 7-1 | CG linear | 99.70 | 99.65 | 1.99 | 70.85 | 30.21 | 100.00 | 0.00 |
|  | CG | 100.00 | 99.92 | 97.39 | 99.99 | 94.09 | 100.00 | 92.89 |

Table 32: Full results of $\beta$-TCVAE on 3dshapes for the compositional generalization setting.

|  |  | floor h | wall h | size | orient | obj h | shape | NC |
|---|---|---|---|---|---|---|---|---|
| NC 0-0 | CG linear | 100 | 99.8 | 0.00 | 65.11 | 47.82 | 48.82 | 0.00 |
|  | CG | 100 | 100 | 38.69 | 100.00 | 40.93 | 99.16 | 0.73 |
| NC 5-2 | CG linear | 93.81 | 95.39 | 0.00 | 59.65 | 56.00 | 77.95 | 0.00 |
|  | CG | 100 | 100 | 0.00 | 99.97 | 100.00 | 100.00 | 0.00 |
| NC 2-1 | CG linear | 88.57 | 90.95 | 0.00 | 33.27 | 2.14 | 61.83 | 0.00 |
|  | CG | - | - | 1.92 | 99.32 | 51.35 | 100.00 | 0.91 |
| NC 3-1 | CG linear | 93.55 | 88.59 | 0.00 | 56.30 | 0.00 | 65.76 | 0.00 |
|  | CG | 100 | 100 | 49.81 | 100.00 | 98.73 | 100.00 | 48.74 |
| NC 3-2 | CG linear | 100 | 100 | 0.00 | 83.40 | 99.06 | 100.00 | 0.00 |
|  | CG | 100 | 100 | 11.51 | 99.80 | 99.90 | 100.00 | 11.51 |
| NC 7-3 | CG linear | 91.25 | 89.19 | 13.79 | 51.55 | 0.00 | 97.49 | 0.00 |
|  | CG | 100 | 95.83 | 26.48 | 94.97 | 67.05 | 100.00 | 2.47 |
| NC 0-3 | CG linear | 98.27 | 99.94 | 14.73 | 58.26 | 61.55 | 90.27 | 0.32 |
|  | CG | 0 | 100 | 22.18 | 99.63 | 56.30 | 99.41 | 0.95 |
| NC 7-1 | CG linear | 100 | 100 | 79.76 | 98.46 | 28.47 | 99.78 | 8.97 |
|  | CG | 100 | 100 | 79.76 | 98.46 | 28.47 | 99.78 | 8.97 |

Table 33: Full results of FactorVAE on 3dshapes for the compositional generalization setting.

|  |  | floor h | wall h | size | orient | obj h | shape | NC |
|---|---|---|---|---|---|---|---|---|
| NC 0-0 | CG linear | 92.95 | 91.58 | 0.75 | 76.03 | 80.19 | 0.00 | 0.43 |
|  | CG | 99.07 | 99.04 | 31.42 | 99.35 | 92.53 | 67.45 | 27.71 |
| NC 5-2 | CG linear | 98.63 | 91.13 | 0.00 | 89.03 | 94.97 | 13.33 | 0.00 |
|  | CG | 0.00 | 99.34 | 2.49 | 99.71 | 99.87 | 99.93 | 2.49 |
| NC 2-1 | CG linear | 91.69 | 93.41 | 57.20 | 85.97 | 97.07 | 0.00 | 55.72 |
|  | CG | 99.87 | 99.12 | 94.90 | 99.71 | 99.57 | 64.74 | 94.53 |
| NC 3-1 | CG linear | 96.72 | 91.91 | 0.00 | 83.17 | 86.87 | 99.97 | 0.00 |
|  | CG | 99.91 | 97.51 | 0.00 | 99.07 | 99.82 | 99.94 | 0.00 |
| NC 3-2 | CG linear | 93.36 | 98.02 | 0.00 | 86.93 | 89.82 | 92.88 | 0.00 |
|  | CG | 99.55 | 99.75 | 1.97 | 99.49 | 99.69 | 99.95 | 1.97 |
| NC 7-3 | CG linear | 94.49 | 89.35 | 0.00 | 67.20 | 71.25 | 0.64 | 0.00 |
|  | CG | 98.38 | 96.93 | 18.09 | 67.31 | 96.62 | 98.57 | 17.62 |
| NC 0-3 | CG linear | 98.17 | 97.79 | 0.00 | 80.65 | 80.35 | 28.11 | 0.00 |
|  | CG | 99.09 | 98.93 | 0.13 | 97.96 | 73.55 | 24.63 | 0.01 |
| NC 7-1 | CG linear | 94.46 | 98.07 | 21.74 | 86.32 | 98.42 | 0.00 | 21.72 |
|  | CG | 98.29 | 99.45 | 28.81 | 98.67 | 99.91 | 85.17 | 28.81 |

Table 34: Full results of PartedVAE on 3dshapes for the compositional generalization setting.

|  |  | floor h | wall h | size | orient | obj h | shape | NC |
|---|---|---|---|---|---|---|---|---|
| NC 0-0 | CG linear | 84.14 | 81.98 | 2.17 | 15.61 | 0.80 | 86.56 | 0.00 |
|  | CG | 98.41 | 98.77 | 23.85 | 20.11 | 71.33 | 97.00 | 13.74 |
| NC 5-2 | CG linear | 97.77 | 97.29 | 0.00 | 88.28 | 5.29 | 91.21 | 0.00 |
|  | CG | 98.98 | 99.91 | 30.91 | 99.23 | 99.21 | 99.17 | 30.91 |
| NC 2-1 | CG linear | 99.96 | 92.79 | 0.00 | 88.49 | 19.61 | 91.82 | 0.00 |
|  | CG | 99.99 | 99.35 | 30.35 | 99.49 | 100.00 | 99.22 | 30.35 |
| NC 3-1 | CG linear | 99.61 | 90.49 | 0.00 | 23.73 | 83.33 | 93.39 | 0.00 |
|  | CG | 99.91 | 99.11 | 38.73 | 73.62 | 98.71 | 99.65 | 38.31 |
| NC 3-2 | CG linear | 93.93 | 98.53 | 6.41 | 38.83 | 99.09 | 94.86 | 5.69 |
|  | CG | 99.87 | 99.36 | 95.13 | 65.39 | 99.93 | 99.45 | 95.06 |
| NC 7-3 | CG linear | 98.71 | 93.34 | 0.00 | 72.27 | 18.47 | 89.05 | 0.00 |
|  | CG | 97.44 | 99.29 | 37.93 | 83.13 | 85.01 | 97.67 | 37.31 |
| NC 0-3 | CG linear | 83.18 | 89.81 | 0.06 | 45.12 | 44.21 | 80.80 | 0.03 |
|  | CG | 83.29 | 99.48 | 4.13 | 57.53 | 23.19 | 81.03 | 1.50 |
| NC 7-1 | CG linear | 70.73 | 83.83 | 45.95 | 13.14 | 21.61 | 84.67 | 2.12 |
|  | CG | 97.48 | 99.06 | 30.89 | 20.65 | 56.57 | 98.29 | 12.87 |

Table 35: Full results of PartedVAE on 3dshapes for the compositional generalization setting.

|  |  | floor h | wall h | size | orient | obj h | shape | NC |
|---|---|---|---|---|---|---|---|---|
| NC 0-0 | CG linear | 72.72 | 76.91 | 6.35 | 47.21 | 82.32 | 65.43 | 2.61 |
|  | CG | 78.71 | 77.17 | 44.06 | 73.61 | 82.43 | 79.39 | 33.00 |
| NC 5-2 | CG linear | 95.11 | 95.78 | 0.00 | 67.89 | 28.85 | 94.55 | 0.00 |
|  | CG | 94.98 | 94.80 | 2.63 | 90.65 | 99.87 | 95.61 | 2.63 |
| NC 2-1 | CG linear | 97.13 | 96.12 | 0.00 | 61.19 | 37.03 | 93.30 | 0.00 |
|  | CG | 97.71 | 96.29 | 19.42 | 89.35 | 97.95 | 96.95 | 19.32 |
| NC 3-1 | CG linear | 93.30 | 94.49 | 0.07 | 93.72 | 88.01 | 91.71 | 0.01 |
|  | CG | 92.23 | 93.37 | 0.00 | 98.49 | 99.52 | 94.33 | 0.00 |
| NC 3-2 | CG linear | 97.22 | 95.55 | 0.00 | 75.38 | 3.08 | 92.79 | 0.00 |
|  | CG | 95.69 | 94.25 | 8.77 | 92.27 | 99.25 | 96.82 | 8.67 |
| NC 7-3 | CG linear | 88.71 | 95.09 | 0.46 | 12.38 | 48.32 | 92.52 | 0.00 |
|  | CG | 87.91 | 95.92 | 6.57 | 68.71 | 100.00 | 94.47 | 6.57 |
| NC 0-3 | CG linear | 97.44 | 94.11 | 6.71 | 17.36 | 59.99 | 88.23 | 0.40 |
|  | CG | 98.04 | 96.01 | 22.81 | 84.21 | 91.52 | 93.67 | 15.16 |
| NC 7-1 | CG linear | 92.85 | 89.84 | 0.61 | 71.59 | 0.28 | 92.05 | 0.00 |
|  | CG | 89.60 | 87.49 | 0.29 | 78.94 | 97.97 | 93.85 | 0.29 |

Table 36: Full results of WeakDE on 3dshapes for the compositional generalization setting.

|  |  | hor | vert | size | obj h | shape | cam he | bg h | NC |
|---|---|---|---|---|---|---|---|---|---|
| NC 0-0 | CG linear | 9.68 | 7.66 | 80.89 | 95.20 | 0.13 | 99.37 | 38.20 | 0.00 |
|  | CG | 75.16 | 55.28 | 99.86 | 98.63 | 20.23 | 100.00 | 100.00 | 20.12 |
| NC 1-4 | CG linear | 16.50 | 6.34 | 42.34 | 91.57 | 5.06 | 99.00 | 99.85 | 0.00 |
|  | CG | 51.51 | 37.19 | 94.21 | 99.93 | 17.16 | 100.00 | 0.00 | 15.56 |
| NC 0-5 | CG linear | 12.55 | 12.39 | 83.41 | 90.54 | 26.06 | 99.98 | 50.46 | 25.62 |
|  | CG | 56.62 | 47.98 | 93.39 | 99.97 | 21.40 | 100.00 | 100.00 | 15.56 |
| NC 1-3 | CG linear | 16.35 | 10.47 | 67.32 | 81.67 | 0.00 | 99.36 | 57.25 | 0.00 |
|  | CG | 87.19 | 65.85 | 99.96 | 99.98 | 0.62 | 100.00 | 100.00 | 0.62 |
| NC 1-1 | CG linear | 15.79 | 10.85 | 92.48 | 74.80 | 0.00 | 97.95 | 55.88 | 0.00 |
|  | CG | 73.84 | 56.43 | 99.98 | 99.77 | 7.93 | 100.00 | 99.70 | 7.91 |
| NC 0-2 | CG linear | 16.22 | 10.56 | 70.52 | 80.97 | 0.00 | 99.94 | 46.48 | 0.00 |
|  | CG | 86.08 | 67.12 | 99.24 | 99.98 | 0.55 | 100.00 | 100.00 | 0.52 |

Table 37: Full results of $\beta$-VAE on MPI3D for the compositional generalization setting.

|  |  | hor | vert | size | obj h | shape | cam he | bg h | NC |
|---|---|---|---|---|---|---|---|---|---|
| NC 0-0 | CG linear | 15.62 | 9.01 | 97.71 | 40.08 | 0.03 | 100.00 | 53.47 | 0.01 |
|  | CG | 69.87 | 53.41 | 99.87 | 96.95 | 2.30 | 100.00 | 100.00 | 2.30 |
| NC 1-4 | CG linear | 18.32 | 12.63 | 68.95 | 72.15 | 1.31 | 99.99 | 45.06 | 0.00 |
|  | CG | 54.29 | 40.48 | 93.83 | 99.66 | 13.79 | 100.00 | 100.00 | 11.74 |
| NC 0-5 | CG linear | 6.51 | 10.99 | 93.31 | 38.70 | 20.30 | 100.00 | 44.35 | 20.30 |
|  | CG | 51.64 | 41.37 | 88.87 | 95.26 | 10.13 | 100.00 | 99.98 | 5.80 |
| NC 1-3 | CG linear | 15.55 | 8.44 | 86.18 | 60.64 | 0.00 | 99.98 | 45.47 | 0.00 |
|  | CG | 85.38 | 64.75 | 99.97 | 99.69 | 0.53 | 100.00 | 98.46 | 0.53 |
| NC 1-1 | CG linear | 12.76 | 8.45 | 78.44 | 74.00 | 0.00 | 99.18 | 44.43 | 0.00 |
|  | CG | 67.06 | 51.68 | 99.90 | 99.53 | 11.52 | 100.00 | 100.00 | 11.47 |
| NC 0-2 | CG linear | 18.69 | 9.13 | 73.26 | 78.03 | 0.00 | 100.00 | 44.74 | 0.00 |
|  | CG | 89.86 | 70.40 | 98.49 | 99.76 | 0.24 | 100.00 | 99.99 | 0.20 |

Table 38: Full results of $\beta$-TCVAE on MPI3D for the compositional generalization setting.

|        |           | hor   | vert  | size  | obj h | shape | cam he | bg h  | NC    |
|--------|-----------|-------|-------|-------|-------|-------|--------|-------|-------|
| NC 0-0 | CG linear | 16.06 | 8.90  | 71.63 | 30.92 | 6.94  | 97.43  | 51.25 | 4.30  |
|        | CG        | 65.00 | 50.01 | 99.57 | 93.45 | 9.92  | 99.95  | 99.92 | 9.71  |
| NC 1-4 | CG linear | 14.03 | 11.09 | 40.42 | 57.51 | 0.05  | 99.86  | 42.14 | 0.00  |
|        | CG        | 46.42 | 34.33 | 93.18 | 98.03 | 29.25 | 100.00 | 99.99 | 26.63 |
| NC 0-5 | CG linear | 11.64 | 7.73  | 92.40 | 64.68 | 8.43  | 99.85  | 52.75 | 8.43  |
|        | CG        | 58.01 | 39.48 | 97.24 | 98.22 | 3.66  | 100.00 | 99.97 | 2.70  |
| NC 1-3 | CG linear | 15.18 | 9.77  | 57.91 | 83.34 | 0.00  | 99.33  | 46.93 | 0.00  |
|        | CG        | 79.48 | 60.41 | 99.89 | 99.94 | 0.55  | 100.00 | 99.99 | 0.55  |
| NC 1-1 | CG linear | 17.13 | 11.69 | 74.08 | 76.06 | 0.00  | 98.12  | 46.27 | 0.00  |
|        | CG        | 58.10 | 47.78 | 98.25 | 98.80 | 10.18 | 100.00 | 99.32 | 9.44  |
| NC 0-2 | CG linear | 16.27 | 11.76 | 67.23 | 80.28 | 0.25  | 99.08  | 48.57 | 0.00  |
|        | CG        | 78.80 | 59.93 | 98.58 | 99.75 | 0.62  | 100.00 | 99.98 | 0.55  |

Table 39: Full results of FactorVAE on MPI3D for the compositional generalization setting.

|        |           | hor   | vert  | size  | obj h | shape | cam he | bg h  | NC    |
|--------|-----------|-------|-------|-------|-------|-------|--------|-------|-------|
| NC 0-0 | CG linear | 9.29  | 2.77  | 93.64 | 23.70 | 100   | 92.70  | 0.00  | 0.00  |
|        | CG        | 21.86 | 18.07 | 98.32 | 70.95 | 100   | 99.57  | 0.06  | 0.06  |
| NC 1-4 | CG linear | 12.39 | 5.12  | 60.38 | 38.19 | 100   | 97.27  | 0.00  | 0.00  |
|        | CG        | 27.80 | 18.19 | 80.16 | 85.51 | 100   | 99.52  | 0.08  | 0.07  |
| NC 0-5 | CG linear | 10.58 | 4.20  | 88.16 | 31.08 | 100   | 99.21  | 0.00  | 0.00  |
|        | CG        | 28.58 | 24.29 | 92.93 | 90.35 | 100   | 99.99  | 0.05  | 0.04  |
| NC 1-3 | CG linear | 10.65 | 4.67  | 90.47 | 31.14 | 100   | 96.15  | 0.00  | 0.00  |
|        | CG        | 28.40 | 17.32 | 97.09 | 86.12 | 100   | 98.58  | 0.95  | 0.85  |
| NC 1-1 | CG linear | 11.27 | 4.79  | 84.14 | 31.40 | 100   | 98.36  | 0.00  | 0.00  |
|        | CG        | 31.14 | 18.25 | 92.75 | 88.23 | 100   | 99.66  | 0.02  | 0.01  |
| NC 0-2 | CG linear | 12.43 | 4.90  | 59.53 | 37.04 | 100   | 97.06  | 0.00  | 0.00  |
|        | CG        | 29.03 | 18.11 | 79.58 | 86.12 | 100   | 99.48  | 0.05  | 0.01  |

Table 40: Full results of PartedVAE on MPI3D for the compositional generalization setting.

|        |           | hor   | vert  | size  | obj h | shape | cam he | bg h  | NC    |
|--------|-----------|-------|-------|-------|-------|-------|--------|-------|-------|
| NC 0-0 | CG linear | 11.81 | 4.95  | 78.04 | 25.77 | 0.00  | 100.00 | 42.80 | 0.00  |
|        | CG        | 23.69 | 20.91 | 97.15 | 76.42 | 0.28  | 100.00 | 51.88 | 0.17  |
| NC 1-4 | CG linear | 12.78 | 5.83  | 50.73 | 44.75 | 0.00  | 100.00 | 46.50 | 0.00  |
|        | CG        | 16.32 | 12.89 | 54.24 | 90.76 | 0.50  | 100.00 | 64.07 | 0.02  |
| NC 0-5 | CG linear | 11.32 | 4.36  | 80.33 | 39.31 | 0.00  | 100.00 | 86.30 | 0.00  |
|        | CG        | 11.32 | 4.36  | 80.33 | 39.31 | 0.00  | 100.00 | 86.30 | 0.00  |
| NC 1-3 | CG linear | 13.23 | 7.50  | 84.65 | 37.04 | 0.00  | 99.98  | 76.83 | 0.00  |
|        | CG        | 51.80 | 29.91 | 98.30 | 93.74 | 0.09  | 100.00 | 98.23 | 0.07  |
| NC 1-1 | CG linear | 10.27 | 5.09  | 74.54 | 30.44 | 0.00  | 96.35  | 56.67 | 0.00  |
|        | CG        | 29.27 | 18.07 | 96.88 | 92.19 | 0.49  | 99.96  | 94.38 | 0.31  |
| NC 0-2 | CG linear | 11.04 | 4.42  | 52.11 | 47.71 | 0.00  | 100.00 | 34.08 | 0.00  |
|        | CG        | 28.44 | 20.42 | 73.13 | 88.58 | 0.04  | 100.00 | 38.63 | 0.01  |

Table 41: Full results of PartedVAE-semisupervised on MPI3D for the compositional generalization setting.

|  |  | hor | vert | size | obj h | shape | cam he | bg h | NC |
|---|---|---|---|---|---|---|---|---|---|
| NC 0-0 | CG linear | 14.27 | 13.70 | 30.99 | 29.20 | 0.02 | 100.00 | 95.09 | 0.00 |
|  | CG | 63.79 | 51.82 | 98.45 | 91.12 | 14.62 | 100.00 | 100.00 | 13.84 |
| NC 1-4 | CG linear | 16.80 | 12.68 | 32.39 | 52.54 | 0.00 | 100.00 | 64.50 | 0.00 |
|  | CG | 30.78 | 18.92 | 87.65 | 97.88 | 3.69 | 100.00 | 100.00 | 3.13 |
| NC 0-5 | CG linear | 11.16 | 12.20 | 59.33 | 35.83 | 0.00 | 100.00 | 79.64 | 0.00 |
|  | CG | 45.44 | 35.85 | 96.86 | 95.43 | 13.44 | 100.00 | 100.00 | 12.95 |
| NC 1-3 | CG linear | 18.41 | 11.59 | 59.82 | 37.31 | 0.00 | 100.00 | 62.56 | 0.00 |
|  | CG | 75.88 | 54.61 | 99.15 | 98.37 | 0.00 | 100.00 | 99.35 | 0.00 |
| NC 1-1 | CG linear | 17.74 | 21.09 | 74.95 | 44.00 | 0.00 | 99.99 | 90.88 | 0.00 |
|  | CG | 70.73 | 61.00 | 99.89 | 97.19 | 11.80 | 100.00 | 99.89 | 11.77 |
| NC 0-2 | CG linear | 20.49 | 19.73 | 47.85 | 61.64 | 0.20 | 100.00 | 91.91 | 0.00 |
|  | CG | 80.54 | 69.87 | 95.28 | 98.97 | 1.17 | 100.00 | 100.00 | 0.92 |

Table 42: Full results of WeakDE on MPI3D for the compositional generalization setting.

|  | x | y | size | orient | shape | zs |
|---|---|---|---|---|---|---|
| NC 5-2 | 77.90 | 77.60 | 6.10 | 3.30 | 33.90 | 2.53 |
| NC 2-1 | 94.10 | 95.80 | 17.80 | 3.10 | 25.60 | 3.79 |
| NC 3-1 | 93.90 | 95.80 | 17.80 | 3.60 | 35.60 | 4.48 |
| NC 3-2 | 88.20 | 88.80 | 3.60 | 3.10 | 38.30 | 0.89 |
| Normal test set 1 | 99.00 | 99.40 | 100.00 | 83.30 | 100.00 | 99.97 |
| Normal test set 2 | 99.00 | 98.60 | 99.80 | 75.00 | 100.00 | 99.80 |
| Normal test set 3 | 98.30 | 98.60 | 99.80 | 75.40 | 100.00 | 99.80 |
| Normal test set 4 | 97.20 | 98.80 | 99.40 | 74.10 | 100.00 | 99.37 |
| Normal test set 5 | 98.70 | 97.40 | 99.60 | 73.00 | 100.00 | 99.58 |

Table 43: Full results of MTD on Dsprites for the compositional generalization setting.

|  | floor h | wall h | size | orient | obj h | shape | zs |
|---|---|---|---|---|---|---|---|
| NC 0-0 | 11.10 | 90.90 | 18.80 | 100.00 | 29.80 | 93.00 | 1.29 |
| NC 5-2 | 8.50 | 92.40 | 4.20 | 100.00 | 26.90 | 94.80 | 0.57 |
| NC 2-1 | 8.20 | 90.50 | 12.20 | 100.00 | 27.80 | 93.90 | 4.25 |
| NC 3-1 | 13.90 | 90.20 | 9.40 | 100.00 | 24.70 | 91.30 | 3.22 |
| NC 3-2 | 10.10 | 96.30 | 10.20 | 100.00 | 24.10 | 94.70 | 2.17 |
| NC 7-3 | 10.09 | 96.02 | 12.79 | 99.94 | 25.03 | 96.35 | 2.00 |
| NC 0-3 | 9.28 | 88.57 | 7.69 | 100.00 | 21.31 | 89.02 | 1.05 |
| NC 7-1 | 9.57 | 96.99 | 15.73 | 100.00 | 23.53 | 96.87 | 0.75 |
| Normal test set 1 | 100.00 | 100.00 | 100.00 | 100.00 | 100.00 | 100.00 | 100.00 |
| Normal test set 2 | 100.00 | 100.00 | 100.00 | 100.00 | 100.00 | 100.00 | 100.00 |
| Normal test set 3 | 9.20 | 100.00 | 100.00 | 100.00 | 100.00 | 100.00 | 100.00 |
| Normal test set 4 | 7.30 | 100.00 | 100.00 | 100.00 | 100.00 | 100.00 | 100.00 |
| Normal test set 5 | 20.00 | 100.00 | 100.00 | 100.00 | 100.00 | 100.00 | 100.00 |

Table 44: Full results of MTD on 3dshapes for the compositional generalization setting.

|                   | hor  | vert | size  | obj h  | shape | cam he | bg h   | zs    |
|-------------------|------|------|-------|--------|-------|--------|--------|-------|
| NC 0-0            | 2.20 | 2.30 | 50.50 | 21.70  | 8.50  | 43.90  | 58.60  | 2.67  |
| NC 1-4            | 2.50 | 0.00 | 96.60 | 53.90  | 11.60 | 99.60  | 100.00 | 11.40 |
| NC 0-5            | 1.79 | 2.44 | 49.55 | 20.93  | 7.95  | 42.79  | 58.67  | 7.36  |
| NC 1-3            | 2.32 | 2.52 | 51.52 | 19.62  | 9.77  | 42.05  | 57.68  | 3.06  |
| NC 1-1            | 2.92 | 2.56 | 42.73 | 19.97  | 8.56  | 43.30  | 58.35  | 2.90  |
| NC 0-2            | 2.39 | 2.52 | 59.98 | 21.82  | 6.46  | 45.27  | 58.73  | 2.68  |
| Normal test set 1 | 2.50 | 2.50 | 99.90 | 100.00 | 96.20 | 100.00 | 100.00 | 96.10 |
| Normal test set 2 | 3.43 | 2.75 | 99.92 | 99.99  | 95.94 | 100.00 | 100.00 | 95.88 |
| Normal test set 3 | 2.48 | 3.00 | 99.92 | 99.99  | 96.43 | 100.00 | 100.00 | 96.37 |
| Normal test set 4 | 2.70 | 2.34 | 99.94 | 99.99  | 95.56 | 100.00 | 100.00 | 95.51 |
| Normal test set 5 | 2.65 | 2.46 | 99.75 | 99.99  | 95.09 | 100.00 | 100.00 | 94.87 |

Table 45: Full results of MTD on MPI3D for the compositional generalization setting.

