# OpenReview forum: "Correcting Flaws in Common Disentanglement Metrics"
_ICLR.cc/2024/Conference — ICLR 2024 Conference Withdrawn Submission_

### Official Review · Reviewer_s8qd · 2023-10-30

**Soundness:** 4 excellent
**Presentation:** 1 poor
**Contribution:** 2 fair
**Rating:** 3
**Confidence:** 2

**Summary:**

The paper examines and identifies shortcomings in existing disentanglement metrics, emphasizing the difference between strong and weak disentanglement, an aspect often overlooked in the literature. To address identified issues, two new metrics, single-neuron classification (SNC) and neuron knockout (NK), are proposed and validated against a novel downstream task of compositional generalization. The work is comprehensive, reviewing and comparing several metrics and discussing practical problems.

**Strengths:**

- Thorough Examination: The paper provides a comprehensive analysis of existing disentanglement metrics, identifying specific flaws and weaknesses.

- Strong vs Weak Disentanglement: A clear discussion and differentiation between strong and weak disentanglement are provided, an essential aspect that is often neglected.

- New Metrics: The introduction of two new metrics aimed at strong disentanglement brings additional value to the paper, addressing the problems identified in existing metrics.

- Downstream Task Validation: The use of a downstream task to validate the proposed metrics is an intelligent and practical approach.

**Weaknesses:**

- Lack of Novelty: Despite the detailed analysis and new metrics, the work might not be considered highly novel. The concepts presented can be seen as incremental improvements over existing ideas.
- Presentation and Structure: The paper's presentation could be improved. It delves into long reflections in different sections, leading to a somewhat confusing structure that may hinder the reader's comprehension.

**Questions:**

- How do the proposed metrics compare with other existing metrics in terms of computational efficiency and scalability?
- Can the authors elaborate on potential applications, other than basic CG, where the new metrics would be particularly beneficial compared to existing metrics?
- What are the authors' thoughts on enhancing the novelty of their work?

---

> ### Author Response · Authors · 2023-11-17
>
> *The concepts presented can be seen as incremental improvements over existing ideas.*
> The two main idea of this paper, neuron-factor alignment and the difference between individual vs collective informativeness, are, to our knowledge, completely novel. We do not know of any existing works to explore these ideas.
>
> *How do the proposed metrics compare with other existing metrics in terms of computational efficiency and scalability?*
> SNC is fast, <1 minute for the entire dataset. NK is slower, around 20minutes for the full dataset. This is still faster than DCI, which can take up to one hour.

---

### Official Review · Reviewer_v9Ai · 2023-10-30

**Soundness:** 3 good
**Presentation:** 3 good
**Contribution:** 2 fair
**Rating:** 3
**Confidence:** 4

**Summary:**

This paper pointed out two problems of existing disentanglement metrics that may lead to incorrect evaluation of certain scenarios, which are (P1) existing metrics fail to penalize bad disentanglement cases that multiple generative factors are learned in the same neuron, and (P2) existing metrics didn't explicitly evaluate the case that one generative factor is learned as some specific functions (e.g. XOR-like function) of multiple neurons. The authors proposed two new disentanglement metrics to overcome these two problems. Experiments on the downstream task of Compositional Generalization demonstrated that the proposed methods achieved a higher correlation with the model's task performance.

**Strengths:**

(1) Evaluating disentanglement is an important topic for representation learning. This paper proposed two new metrics to solve the weakness of existing metrics, which aligns with the interests of the ICLR community.

(2) The findings of (P2) are novel and very interesting, which can share light and benefit future research on understanding the disentanglement of generative factors with respect to combinational effects of multiple latent variables, and help the discussion of strong disentanglement vs weak disentanglement.

(3) Experiments (especially Table 3) show the proposed metrics are promising for selecting good disentangled models for downstream tasks related to compositional generalization.

(4) Overall, the paper is well-structured and easy to follow.

**Weaknesses:**

(1) A great number of related works that are important for (P1) are overlooked. If I understand correctly, the authors pointed out that existing works like MIG fail to penalize learning multiple generative factors in the same neuron, which is true. However, there are a lot of other existing metrics that are designed exactly to measure this, for example, MIG-sup [1], modularity metrics (Eqn (2) in [2]), and more that are reviewed in [3] in the category of "Modularity". The authors are encouraged to discuss and compare these metrics.

(2) This paper didn't clearly demonstrate how the proposed metrics solve the proposed problems. The authors are encouraged to design experiments that can clearly demonstrate that the proposed metrics can correctly evaluate the scenarios of (P1) and (P2), while the existing metrics, on the other hand, may fail and produce a "false" high disentanglement score.

[1] Li, Zhiyuan, et al. "Progressive learning and disentanglement of hierarchical representations." ICLR 2020.
[2]
Ridgeway, Karl, and Michael C. Mozer. "Learning deep disentangled embeddings with the f-statistic loss." Advances in neural information processing systems 31 (2018).
[3] Carbonneau, Marc-André, et al. "Measuring disentanglement: A review of metrics." IEEE transactions on neural networks and learning systems (2022).

**Questions:**

(1) It would be appreciated if the authors could provide clarifications and responses to the concerns raised in the "Weaknesses" section, especially weakness (1).

(2) Does the two proposed metrics agree with each other well? Because big differences between them are observed in some cells in Table 1.

(3) What perspective of disentanglement does the two proposed metrics are focused on? For example, using the terms "modularity", "compactness", and "explicitness" in [2] or universal?

---

> ### Author Response · Authors · 2023-11-17
>
> *Discussion of MIG-sup*.
> Thank for this reference, we will include it in the discussion of single neurons representing multiple factors. It is indeed a solution to this problem, but we would argue it is an imperfect solution. In the comparison between the two toy models in Section 3.1, M1 would get a higher MIG score but a lower MIG-sup, and the sum MIG+MIG-sup would be the same for both models. For our metrics, however, the sum SNC+NK would be lower, which we have argued to be more correct. There is also the problem that it is not clear how to extend MIG and MIG-sup to deal with P2, because extending MI to multiple variables becomes tricky both theoretically and computationally.
>
> Modularity in [2] takes a similar solution to this problem, similar also to DCI, though like DCI, we do not claim that this problem occurs in [2] because it doesn't operate by explicitly aligning factors to neurons, as MIG does. Also, all three of these approaches, DCI, MIG and [2], penalize a neuron representing multiple factors by adding another metric. Our work instead employs the cleaner solution of forcing the alignment to be injective in the first place.
>
> *The authors are encouraged to design experiments that can clearly demonstrate that the proposed metrics can correctly evaluate the scenarios of (P1) and (P2), while the existing metrics, on the other hand, may fail and produce a "false" high disentanglement score.*
> The solution of P1 is automatic, by the definition of our metrics. For P2, it would be straightforward to show that NK will pick up on $z_{\neq i}$ representing $z_i$ by XOR or something similar, as learning a function like XOR is a simple task for an MLP. We will add an experiment of this sort.
>
> *What perspective of disentanglement does the two proposed metrics are focused on? For example, using the terms "modularity", "compactness", and "explicitness" in [2] or universal?*
> They do not fit so neatly into the framework of [2]. NK corresponds to modularity, but SNC is a mixture of explicitness and compactness.

---

> > ### Comment · Reviewer_v9Ai · 2023-11-22
> >
> > Thank the authors for their response. I will keep my score. More experiment results (as suggested in Weakness (2)) are necessary to support the authors' claim that the proposed methods are more correct to the existing imperfect solutions.

---

### Official Review · Reviewer_zKra · 2023-10-31

**Soundness:** 2 fair
**Presentation:** 2 fair
**Contribution:** 1 poor
**Rating:** 5
**Confidence:** 3

**Summary:**

The paper is focused on the correction of flaws in common disentanglement metrics used in learning disentangled representations. The authors identify two shortcomings in existing metrics:1) incorrect assignment of factors to the most informative neuron. 2) focusing on single neurons instead of multiple neurons. To overcome these limitations, the authors propose two new metrics SNC and NK. SNC calculates the accuracy of the single-neuron classifier and the NK is the accuracy difference of knocking one latent variable. Then, the authors demonstrate the results on Dsprites, 3dshapes, and mpi3d to demonstrate the superiority of the proposed metrics using compositional generation tasks.

**Strengths:**

1. It clearly identifies limitations of existing disentanglement metrics through theoretical analysis and empirical demonstrations. The flaws pointed out, such as incorrect neuron-factor alignment and ignoring distributed representations, are insightful and not obvious initially.

2. Theoretical analysis via the proof of issues with DCI metric demonstrates rigor.

**Weaknesses:**

1. The validation of proposed metrics is poor. The authors should run experiment on some artificial models to prove the effectiveness of these metrics by deliberately creating some entangled representations, like Figure 4 in [1].
2. The proposed metrics seem having flaws too. Consider that there are two factors and  z1 = c1 + c2 and z_i≠1 = 0, in this case two factors are represented by z1. The SNC will be zero. Though it is arguable about the definition of disentanglement, this toy example should be the worst case.
3. The example in Figure 1 right is not clear. What is the coding map for z1? I suggest the authors to show the map between factors and the code in a table, like (square,blue)→0.


Reference:
[1] Jiantao Wu, Shentong Mo, and Lin Wang. 2021. An Empirical Study of Uncertainty Gap for Disentangling Factors. In Proceedings of the 1st International Workshop on Trustworthy AI for Multimedia Computing (Trustworthy AI'21). Association for Computing Machinery, New York, NY, USA, 1–8. https://doi.org/10.1145/3475731.3484954

**Questions:**

Suggestions:
1. A proper order in Related Work can be Definitions, Model and Metric.
2. The second metric is not clear. It should be something like MEAN(NKi) instead of NKi

---

> ### Author Response · Authors · 2023-11-17
>
> Thank you for your time and comments. Please see our replies below.
>
> *Consider that there are two factors and z1 = c1 + c2 and z_i≠1 = 0, in this case two factors are represented by z1. The SNC will be zero. Though it is arguable about the definition of disentanglement, this toy example should be the worst case.*
> SNC will not be zero, because a higher value of $z_1$ will make a higher value of $c_1$ more likely, and the same for $c_2$. It will be low, but that is correct this case. Whatever definition of DE one has in mind, this scenario clearly is not disentangled, as each neuron is only supposed to represent a single factor, and represent it very accurately, whereas here, $z_1$ represents two factors, both imperfectly.
>
> *The example in Figure 1 right is not clear. What is the coding map for z1? I suggest the authors to show the map between factors and the code in a table, like (square,blue)→0.*
> Thank you. We can reformat as you suggest.

---

### Official Review · Reviewer_ok83 · 2023-11-08

**Soundness:** 3 good
**Presentation:** 3 good
**Contribution:** 2 fair
**Rating:** 3
**Confidence:** 3

**Summary:**

The authors propose two metrics for evaluating disentangled representations. More specifically they formulate the notion of hard disentanglement where each neuron (or axis) represents a single generating factor in the data. They argue and present empirical results that existing disentanglement metrics fail to correctly estimate the amount of disentanglement. They introduce two new metrics, single-neuron classification and neuron knockout that overcome the limitations of the existing metrics on the hard disentanglement task.
They validate their approach on the task of combinatorial generalization using three well-established disentanglement datasets and seven disentanglement approaches and variants.

**Strengths:**

The paper is written well and clearly identifies the limitations of the existing metrics for the given goal. A good background is given on the existing metrics and a clear overall discussion on disentanglement and its goals.

The two proposed methods are clearly described.

The empirical analysis covers a sufficiently broad range of methods and metrics.

**Weaknesses:**

Even though disentangling each generating factor to a different neuron in the representation may seem a clearly well-formulated goal, usually from the perspective of interpretability or combinatorial generalization, it also may be an unnecessarily high bar to expect from machine learning models. There has been a lot of criticism towards the field of learning disentangled representation, particularly in an unsupervised fashion that aligns with having this high expectation. The authors did recognize this discussion in the related work section, mentioning the work of Locatelo at al. But they quickly dismiss this by referencing two other papers and do not go much into this discussion.

However, given that this paper proposes two new metrics motivated with fairly straightforward intuition, but not a method that would actually deliver hard disentanglement, the main weakness becomes the contribution of this work itself. There may be a core incompatibility between how (deep) neural networks learn distributed representations, particularly the need to overparameterize to be able to train such models and the goal of aligning a single factor to a single neuron.

Without a stronger validation of the feasibility of training hard disentangled representation, the new metrics seem as a solution to a problem that has a limited impact.

**Questions:**

In terms of the formulation of disentanglement. How are the generating factors defined? As with any data description, these factors may have various levels of semantic abstraction. So, defining them precisely seems hard to do.

Are there disentanglement methods that can achieve high disentanglement levels on a broad range of disentanglement tasks with the proposed metrics?

If so, what are the limitations of these methods?

---

> ### Author Response · Authors · 2023-11-17
>
> Thank you for your time and your review. Please see our replies below.
>
> *Strong DE may be too high a bar.*
> Many existing works still target strong DE and discuss the benefits of their work wrt strong DE.
>
> *The authors dismiss the work of Locatello et. al.*
> The main conclusion of Locatello et al. is that DE requires some training labels. We make clear our view, like that of the other recent works we cite, that, in practice, the work of Locatello is relevant but does not completely preclude work on unsupervised DE. In any case, we also include two semi-supervised models in our experiments.
>
> *In terms of the formulation of disentanglement. How are the generating factors defined?*
> They are the labels in the datasets we use.
>
> *Are there disentanglement methods that can achieve high disentanglement levels on a broad range of disentanglement tasks with the proposed metrics?*
> We are not aware of any tasks that are regarded as DE tasks. If you would like to clarify what tasks you are referring to, we will do our best to answer.

---

### Official Review · Reviewer_cX7W · 2023-11-08

**Soundness:** 2 fair
**Presentation:** 2 fair
**Contribution:** 2 fair
**Rating:** 3
**Confidence:** 4

**Summary:**

This paper proposes 2 new supervised disentanglement metrics, which improve on existing ones. The first metric forces a Hungarian match between factors and latents before computing single latent accuracies. The second one computes a difference between the accuracy of a MLP classifier having access to all latents vs all but the i-th latent.

They show how their metrics fare on the usual trivial/toy datasets used for disentanglement, and also assess the ability of models’ encoders to generalize to held-out transfer inputs. They also show correlations of metrics to this measure of generalisation.

Overall, the proposed metrics are fine, but are not ground-breaking. People used Hungarian matching before (albeit not in metrics, by choice, it comes with an effective metric see below), and using classifiers is also quite standard (although again, not in exactly the same way they do). However some claims in the paper are not consistent with observed evidence and it is unclear how valuable yet another set of disentanglement metrics really are (no evidence of it being useful to do model selection for more complex downstream tasks are provided). This paper is probably not of a large enough scope for ICLR.

**Strengths:**

1. The alignment problem is well defined, and even though I am not sure I would agree with the decision, the choice of going for strong disentanglement is well put forward.
2. The metrics are fine (up to questions below), and I would expect them to improve on the status quo (which is not satisfactory to any practitioner, metrics are hard), although I am not sure supervised metrics is really what is missing right now.

**Weaknesses:**

1. The paper spends a lot of time presenting the problem, even though these are quite known/trivial to the community and these flaws are mostly the effect of historical drift / evolution.
2. In my opinion, strong disentanglement is not something that is of high value to target anymore, and hence the paper is solving for something that the community has moved on from already. This will show up in subtle ways in the metrics, for example how would you expect “color” to be handled by your metrics? What about if your groundtruth labels for generative factors use RGB but the latents use something closer to HSV? Again, strong disentanglement is an odd (historical) target.
3. The presentation and clarity could be improved. A lot of time is spent on setting up the problem and what is wrong, but very little to actually present clearly how the metrics are computed and what matters. Several captions/descriptions refer to missing content (or that is only in the Appendix), and tables use bolding in a confusing way. Overall it needs work.
4. The compositional generalisation section makes some assumptions that are too simplistic and the provided interpretation is not taking into account some existing subtle observations from previous work.
5. It is of limited scope, and it is not clear if this will be useful to the community at large. The correlations are only performed between their own measurements, which could have cyclical dependencies.

**Questions:**

1. How would your metric handle situations which are not compatible with strong-disentanglement?
   1. E.g. the color example above? This happens a lot and in my experience is what makes current metrics “bad”, not just the alignment issue.
2. The Hungarian algorithm usually uses L2 over whichever values you use. You chose MI, which means you’re taking squared distances of bits in order to do tie-breaking or fine-grained factor-latent attribution. Is that really what you want?
   1. This is actually a strong hidden assumption in your metric. In my experience L2 is not always what you need, and situations where multiple alternatives are equally likely do not always behave well without careful consideration.
   2. See for example AlignNet [1] and especially OAT [2] for a more careful discussion of why learning the metric to align on might be necessary in more complex situations.
3. The presentation of the SNC metric is not as clear as it should be.
   1. What is z_i? I understand it’s the output of the hungarian matching from G to Z, but you do not define it
   2. The description of the binning is confusing.
   3. What happens if you had continuous factors/latents?
   4. How does this differ from computing the Discrete mutual information?
4. The NK metric reliance on classifiers might be problematic:
   1. How do you avoid issues with overfitting by using classifiers?
   2. This is partially why DCI used linear classifiers. I am not sure the MLP results in Table 1 are encouraging.
   3. How different is it to doing the “difference between best and next best” latents that most other metric do? Your Hinton diagrams do not have a lot of mass on more than 1-2 factors per latent (as expected if the methods work), does it really matter?
5. Table 1 is not discussed that precisely and Table 4 in the Appendix seems more interesting
   1. Why did you bold all MLP values, instead of bolding across rows to indicate the best model? This was confusing / unhelpful.
   2. Why do you use an MLP despite the issues you raised? How do you counteract overfitting?
   3. The comparison of accuracies at the end of Section 5.1 just does not feel appropriate. You cannot make judgements about linearity of sum of classifier scores vs classifier score of more latents, especially given you used a nonlinear classifier? It also does not bring much.
6. Table 2 cites a CG linear which does not exist.
7. The compositional generalization does not feel consistent with previous work or my own experience with these models. It is both more complex and simpler than you assume.
   1. These models will encode what data they have seen. The fact that they can extrapolate to held-out value combinations depends entirely on the support you chose to train on.
      1. See the Appendix of Spatial Broadcast Decoder [3] for an example of this in action in a simplified setting, compare Figures 13, 14, 15 and 18.
      2. As you can see, having no support in the “middle” of the support is not hurting the generalisation ability of the encoder. It perfectly handles unseen values.
      3. Instead, having values held out at the “edge” of the distribution can result in various failures / under-optimal solutions. The latent space in Figure 15 (say FactorVAE with Spatial Broadcast) does somehow complete the missing corner, but not perfectly. So yes it is not perfect but it also is better than a random manifold?
   2. Your usage of a MLP, especially without careful regularisation, will definitely amplify this effect. Also if you use discrete generative factors, you should not really expect meaningful rank-preserving latents to appear. As you chose size and shape, you are exactly in the regime where no linear extrapolation is likely, so all bets are off (especially on these toy datasets).
   3. In my experience it is definitely possible for these models to extrapolate compositionally, but one has to be extremely careful about it, and they are not magical. But it is not as clear cut a “no” as you assert however.
8. Some examples are contrived and less likely to happen in reality.
   1. For example Section 3.2 proposes a XOR latent to indicate how the metric might fail. How likely is that to happen in practice?

References
- [1] https://arxiv.org/abs/2007.08973
- [2] https://arxiv.org/abs/2103.04693
- [3] https://arxiv.org/abs/1901.07017

---

> ### Author Response · Authors · 2023-11-17
>
> Thank you for your time to read and your comments. Please find our answers below.
>
> *How would your metric handle situations which are not compatible with strong-disentanglement?*
> They are designed for strong DE only. As we argue, one cannot have a single metric that properly measures both weak and strong DE.
>
> *The Hungarian algorithm usually uses L2 over whichever values you use. You chose MI, which means you’re taking squared distances of bits in order to do tie-breaking or fine-grained factor-latent attribution. Is that really what you want?*
> There is no L2 in the Hungarian algorithm. It simply receives as input an $M \times N$ cost matrix $C$, $M \geq N$, and returns some permutation $s \in S_N$ that minimizes $\sum_{i=1}^N C[S_N[i],i]$ (use python-style indexing). There is no mention of L2 in the original papers by Kuhn and Munkres [1,2], nor the documentation for the scipy implementation, which is what we use. The values of our cost matrix are MI, just as we describe, with no notion of squared distances of bits.
>
> *Presentation of metrics, what is $z_i$?*
> Thanks. We will specify that we mean the latent indices are ordered so that $z_i$ corresponds to $g_i$ for each $i$.
>
> *What happens if you had continuous factors/latents?*
> The latents are always continuous. If the factors were continuous, they could be binned in the same way that we bin the latents. Indeed, many of the labels in these datasets essentially are binned continuous values.
>
> *How does SNC differ from discrete mutual information?*
> Both accuracy and MI are roughly used for the same purpose as performance metrics, but they are computed quite differently and are different theoretical quantities. SNC is also chance adjusted.
>
> *How do you avoid issues with overfitting by using classifiers?*
> Note that there is a train-test split used for the MLP classifiers. We employ early-stopping and generally do not observe any overfitting. Note also that most current uses of DCI use a gradient-boosted tree.
>
> *How different is it to doing the “difference between best and next best” latents that most other metric do? Your Hinton diagrams do not have a lot of mass on more than 1-2 factors per latent (as expected if the methods work), does it really matter?*
> Yes, it definitely matters. This is an essential part of our paper which we discussed at length. Having low MI with every neuron individually does not mean that there is low MI with the entire set of neurons. With all due respect, you criticize the paper because we spend too long describing the problems, but you seem to have misunderstood this problem despite our lengthy description.
>
> *Table 4 is more interesting than Table 1.*
> The only difference is that Table 4 has results for a linear classifier as well. We can add this to the main paper.
>
> *Bold the best model in each row*
> Thanks, we will do this.
>
> *Table 2 cites a CG linear which does not exist.*
> Thanks, we will fix this.
>
> *The comparison of accuracies at the end of Section 5.1 just does not feel appropriate.*
> The purpose is not to imply that the collective accuracy and the sum of individual are calibrated on the same scale, but rather to show how the informativeness of each individual neuron can be low, while that of the set is high. This is also the point of the XOR example. XOR itself may not be likely in practice, but the pattern of low-individual-high-collective is.
>
> *Compositional generalization results.*
> We do not claim that these models completely fail at CG. The performance is low, which we believe is supported by our results and those of previous works, but it is not zero. There is two full pages of discussion on the CG results, and we explicitly state, at the bottom of p8 "some models achieve up to 30% even after chance adjusting, which is significantly higher than random guessing". We do not assert a clear-cut no, as you claim.
>
>
> [1] Harold W. Kuhn, "The Hungarian Method for the assignment problem", Naval Research Logistics Quarterly, 2: 83–97, 1955
>
> [2] J. Munkres, "Algorithms for the Assignment and Transportation Problems", Journal of the Society for Industrial and Applied Mathematics, 5(1):32–38, 1957

---

### Meta-Review · Area_Chair_SZCw · 2023-12-01

**Metareview:**

This paper proposes two new supervised disentanglement metrics. Single-neuron Classification (SNC) derives a score from the predictiveness of individual neurons after matching neurons and factors. Neuron Knockout compares the accuracy of predicting a factor from all neurons to the same setting but with one neuron removed. It is shown how these metrics can be used to analyze models on standard benchmarks and how they correlates with model performance on held-out combinations of attributes (compositional generalization).

Most reviewers agree that this paper addresses a well-defined problem and contributes a reasonable (albeit rather incremental) approach that innovates over prior metrics. However, there is broad agreement that problem itself, i.e. that of analyzing strong-disentanglement in neural networks, is of limited significance. Indeed, as reviewer cX7W puts it "the paper is solving for something that the community has moved on from already" or in reviewers ok83's words "Without a stronger validation of the feasibility of training hard disentangled representation, the new metrics seem as a solution to a problem that has a limited impact.". Other considerable weaknesses were also identified, such as limited comparison to related metrics, and concerns regarding the compositional generalization experiments.

While I acknowledge that it is difficult for the authors to address the issues that were brought up regarding significance, this is ultimately an important criteria along which to judge the merit of a paper. On the other hand, the author response could do a better job at convincing the reviewers of why hard disentanglement is an important problem. I would encourage the authors to consider TMLR or a comparable venue that is more lenient in this regard.

**Justification For Why Not Higher Score:**

The main justification is the limited significance of the problem that is being addressed as also pointed out by multiple reviewers.

**Justification For Why Not Lower Score:**

N/A

---

### Decision · Program_Chairs · 2024-01-16

Reject